# Dendrogenin A drives LXR to trigger lethal autophagy in cancers

Gregory Segala[1,2], Marion David[1,3], Philippe de Medina[4], Mathias C. Poirot[1], Nizar Serhan[1], François Vergez[5], Aurelie Mougel[1], Estelle Saland[3], Kevin Carayon[1], Julie Leignadier[1], Nicolas Caron[4], Maud Voisin[1], Julia Cherier[1], Laetitia Ligat[6], Frederic Lopez[6], Emmanuel Noguer[4], Arnaud Rives[4], Bruno Payré[7], Talal al Saati[8], Antonin Lamaziere[9], Gaëtan Despres[9], Jean-Marc Lobaccaro [10], Silvere Baron[10], Cecile Demur[5], Fabienne de Toni[3], Clément Larrue[3], Helena Boutzen[3], Fabienne Thomas[11], Jean-Emmanuel Sarry[3], Marie Tosolini [6], Didier Picard[2], Michel Record[1], Christian Récher[3,5], Marc Poirot [1] & Sandrine Silvente-Poirot[1]

Dendrogenin A (DDA) is a newly discovered cholesterol metabolite with tumor suppressor properties. Here, we explored its efficacy and mechanism of cell death in melanoma and acute myeloid leukemia (AML). We found that DDA induced lethal autophagy in vitro and in vivo, including primary AML patient samples, independently of melanoma Braf status or AML molecular and cytogenetic classifications. DDA is a partial agonist on liver-X-receptor (LXR) increasing Nur77, Nor1, and LC3 expression leading to autolysosome formation. Moreover, DDA inhibited the cholesterol biosynthesizing enzyme 3β-hydroxysterol-$\Delta^{8,7}$-isomerase (D8D7I) leading to sterol accumulation and cooperating in autophagy induction. This mechanism of death was not observed with other LXR ligands or D8D7I inhibitors establishing DDA selectivity. The potent anti-tumor activity of DDA, its original mechanism of action and its low toxicity support its clinical evaluation. More generally, this study reveals that DDA can direct control a nuclear receptor to trigger lethal autophagy in cancers.

[1] UMR 1037-CRCT, Université de Toulouse, INSERM, UPS, Cholesterol Metabolism and Therapeutic Innovations Team, Toulouse, F-31037, France. [2] Département de Biologie Cellulaire, Université de Genève, Genève, 1211, Switzerland. [3] UMR 1037-CRCT, Université de Toulouse, INSERM, UPS, Chemoresistance, Stem Cells and Metabolism in Acute Myeloid Leukemia, Toulouse, F-31037, France. [4] AFFICHEM, Toulouse, F-31400, France. [5] Service d'Hématologie, Institut Universitaire du Cancer de Toulouse-Oncopole, CHU de Toulouse, Toulouse, F-31100, France. [6] UMR 1037-CRCT, Pole Technologique, Toulouse, F-31037, France. [7] Centre de Microscopie Electronique Appliquée à la Biologie, Toulouse, F-31062, France. [8] INSERM-US006 ANEXPLO/CREFRE F-31024, Toulouse, F-31024, France. [9] Laboratory of Mass Spectrometry, INSERM ERL 1157, CNRS UMR 7203 LBM, Sorbonne Universités-UPMC, CHU Saint-Antoine, Paris, F-75012, France. [10] Université de Clermont Auvergne, CNRS, INSERM, GReD, Clermont-Ferrand, F-63001, France. [11] UMR 1037-CRCT, Université de Toulouse, INSERM, UPS, Dose Individualisation of Anticancer Drugs Team, Toulouse, F-31037, France. Gregory Segala, Marion David, and Philippe de Medina contributed equally to this work. Correspondence and requests for materials should be addressed to C.R. (email: recher.christian@iuct-oncopole.fr) or to M.P. (email: marc.poirot@inserm.fr) or to S.S.-P. (email: sandrine.poirot@inserm.fr)

Deregulation at various points along the cholesterol metabolic pathway has recently been shown to favor the accumulation of metabolites with tumor-promoting activity[1–4], however a cholesterol metabolite was also discovered in human tissues and cells, named dendrogenin A (DDA), with anti-tumor properties[4–8]. In vitro, DDA triggers cancer cell differentiation and death[9]. In vivo, DDA controls the growth of mouse tumors and increases animal survival and these effects were associated with tumor differentiation and cholesterol epoxide hydrolase (ChEH) inhibition[5]. Interestingly, DDA levels were decreased in patient tumors and it was not detected in a panel of cancer cell lines, suggesting a deregulation of DDA biosynthesis during carcinogenesis and a physiological function in maintaining cell integrity[5]. Thus, DDA appears to be the first tumor suppressor of cholesterol origin discovered so far with potential clinical interest[2]. However, its efficacy in vivo against human tumors and the mechanisms involved in its anticancer activity have not yet been evaluated. ChEH activity is carried out by two enzymatic subunits, the 3β-hydroxysterol-$\Delta^{8,7}$-isomerase (D8D7I or EBP) and 3β-hydroxysterol-$\Delta^7$-reductase (DHCR7)[10], which are both involved in cholesterogenesis. ChEH inhibitors such as the anticancer drug Tamoxifen (Tam), have been shown to induce tumor cell differentiation and death and survival macroautophagy (hereafter referred as to autophagy)[11–16]. Cell differentiation and death was due to the cholesterol epoxides accumulation through the stimulation of cholesterol epoxidation and the inhibition of ChEH[11, 12, 17]. Autophagy induced by Tam and selective ChEH inhibitors such as PBPE has been associated with the accumulation of free sterols due to the inhibition of D8D7I[15]. It is a physiological process that maintains homeostatic functions and cell survival. Cancers can upregulate autophagy to survive microenvironmental stress and to increase growth and aggressiveness[18]. However, recent data have provided evidence that the autophagic machinery can also be recruited to mediate selective tumor cell death, anti-tumor immunity and can be crucial for vital functions such as developmental morphogenesis, tissue homeostasis and the counteraction of aberrant cell division[19–22].

In the present study, we report the potent anti-tumor activity of DDA against human melanoma and acute myeloid leukemia (AML) both in vitro and in vivo, including primary tumors from AML patients. Further, we describe its original mechanism of cytotoxicity, which involves the direct control of a nuclear receptor to trigger lethal autophagy.

## Results

### DDA induces melanoma cell death independent of apoptosis.

In murine B16F10 and human SKMEL-28 melanoma cells, DDA (Fig. 1a) induced cytotoxicity and inhibited clonogenicity while its regio-isomer C17 (Fig. 1a) was inactive (Fig. 1b; Supplementary Fig. 1a). Sensitivity to DDA was also observed in various human melanoma cell lines irrespective of their Braf status (Supplementary Fig. 1b). In the melanoma cell lines B16F10 and SKMEL-28, DDA induced tumor cell accumulation in sub G0/G1, and the appearance of characteristics of apoptosis (Supplementary Fig. 1c–g), however DDA cytotoxicity measured for 48 and 72 h was not blocked by general caspase inhibitors or antioxidants which blocked lipoperoxidation and cholesterol epoxidation (Fig. 1c), suggesting that cell death is independent of apoptosis and ChEH inhibition. Analyses of the oxysterol profile of cells treated with DDA showed no accumulation in 5,6-EC as opposed to what was found with other ChEH inhibitors Tam and PBPE (Supplementary Fig. 1h). We observed that DDA stimulated catalase activity (Supplementary Fig. 1i), which induced $H_2O_2$ destruction and blocked 5,6-EC production. This established a

significant difference between DDA and ChEH inhibitors like Tam or PBPE (Supplementary Fig. 1j, k) because we showed that Tam and PBPE mediated part of their cytotoxicity through the accumulation of 5,6α-EC, which acted as a second messenger[17]. DDA cytotoxicity was inhibited by actinomycin D and cycloheximide, indicating that cell death triggered by DDA required gene transcription and translation (Fig. 1c). Inhibition of one of the ChEH subunit, D8D7I, and the accumulation of its substrate, zymostenol, has been previously reported to be associated with autophagy[11, 14, 15, 23]. Here we found that DDA inhibited D8D7I and induced the accumulation of Δ8-sterols (zymostenol and 8-dehydrocholesterol) (Supplementary Fig. 2a), illustrated by the appearance of intracellular filipin punctate labeling of free sterols (Fig. 1d). This showed that the consequence of DDA binding on ChEH is the inhibition of D8D7I and Δ8-sterols accumulation but not 5,6-EC accumulation as observed with other ChEH inhibitors such as tamoxifen[17] (Supplementary Fig. 2b–d). Ultrastructure analysis confirmed that DDA-treated cells contained white cytosolic vesicles (Supplementary Fig. 2e: panels 3, 4), which were identified as lysosomes (Ly), autophagosomes (AP) and autolysosomes (AL) (Fig. 1e: panels 1, 2 and 3, 4 respectively). Numerous filipin punctates, marking the sites of Δ8-sterols accumulation, were also found to express the lysosomal marker LAMP1 (Supplementary Fig. 2f), showing their lysosomal nature and corresponded to the multilamellar bodies[16] observed by transmission electron microscopy (TEM) (Fig. 1e). Collectively, these data indicate that DDA induces a caspase- and reactive oxygen species-independent cell death and characteristics of autophagy.

### DDA induces Nur77- and NOR1-dependent lethal autophagy.

As DDA-mediated melanoma cell death was found to require gene transcription, we measured the effect of DDA treatment on the transcription of genes encoding nuclear receptors (NR) and their co-regulators, using PCR arrays. DDA stimulated the transcription (Fig. 1f) and protein expression (Fig. 1g) of the transcription factors NR4A1 (Nur77) and NR4A3 (NOR1). Accordingly, single or double knockdown (KD) of Nur77 and NOR1 impaired DDA cytotoxicity (Fig. 1h; Supplementary Fig. 2g). These data indicate that Nur77 and NOR1 are involved in DDA-induced cell death. To validate the induction of autophagy suggested by TEM analysis, complimentary assays were performed[23, 24]. DDA induced the accumulation of monodansylcadaverine (MDC)-positive vesicles (Fig. 2a), which stains Ly, AL, and AP[25]. It also increased levels of autophagic proteins, with a main impact on LC3-II expression (Fig. 2b), and stimulated the rate of long-lived protein degradation following its inhibition by two pharmacological inhibitors of autophagy, bafilomycin A1 (Baf A1) and hydroxychloroquine (HCQ) (Fig. 2c). This confirms the functional induction of autophagy by DDA. DDA was also shown to induce autophagic flux as the co-treatment of DDA with lysosomal degradation inhibitors (E64d and pepstatin A) further enhanced (~2-fold) LC3-II accumulation (Fig. 2d) and GFP-LC3 puncta (Fig. 2e) relative to DDA alone. The lysosomal inhibitor Baf A1 inhibited the production of MDC-positive vesicles induced by DDA (Supplementary Fig. 2h) and, in line with this, KD of the autophagic proteins ATG7, VPS34, and BECN1 in melanoma cells (Supplementary Fig. 2i) inhibited DDA-induced LC3-II formation (Supplementary Fig. 2j), DDA-induced MDC-labeled punctates (Supplementary Fig. 2k), and DDA-induced cell death (Fig. 2f–g). Blocking autophagy with Baf A1 or HCQ also inhibited DDA-induced cell death (Fig. 2h). In contrast, Baf A1 or HCQ treatment potentiated the toxicity induced by inhibitors of D8D7I[26] such as PBPE and Tam (Supplementary Fig. 3a, b), which induced a protective

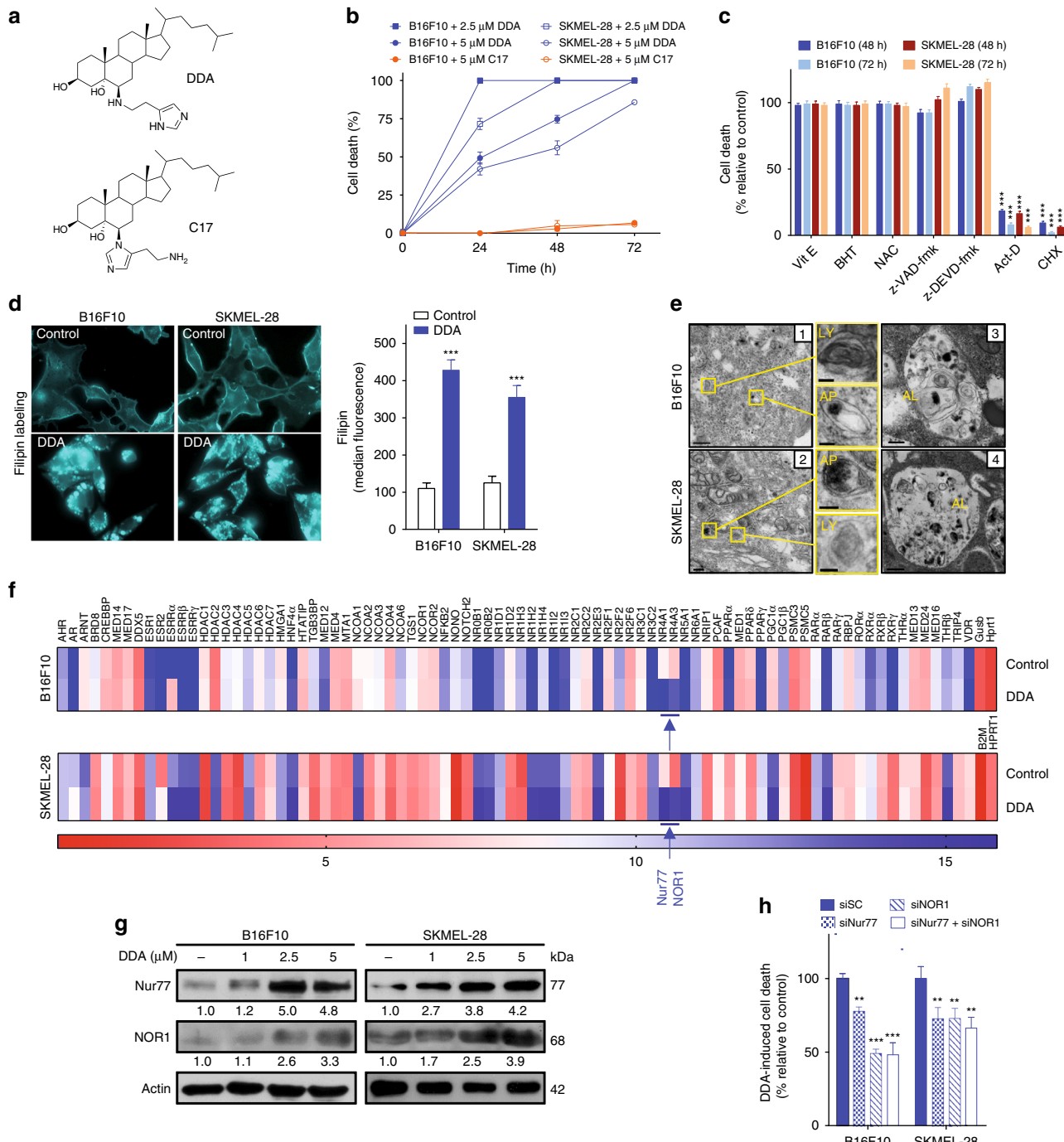

**Fig. 1** Nur77 and NOR1 are required for the induction of cell death by DDA in melanoma. **a** Chemical structure of DDA and C17. **b** B16F10 and SKMEL-28 cells were treated or not for 24, 48, or 72 h with DDA or C17. DDA- or C17-induced cell death was determined by a trypan blue assay and expressed as the percentage cell death relative to control (vehicle). **c** Cells were treated with 5 μM DDA for 48 h or 72 h with or without 500 μM vitamin E (Vit E), 50 μM z-VAD-fmk, 50 μM z-DEVD-fmk, 1 μg/ml actinomycin D (Act D) or 2.5 μg/ml cycloheximide (CHX), and cell death was measured as in **b**. **d** DDA induced the accumulation of free sterols in cells. Cells were treated with solvent vehicle or 2.5 μM DDA for 48 h, then fixed and stained with filipin and analyzed by fluorescence microscopy. **e** Representative EM images of B16F10 (panels 1–3) and SKMEL-28 (panels 2–4) cells treated for 24 h with 2.5 μM DDA. Ly: multilamellar body-derived lysosomes, AP: autophagosomes, AL: autolysosomes, N: nucleus, C: cytoplasm. Bars: 250 nm for panel 1 and 100 nm for inserts; 500 nm for panel 2 and 100 nm for inserts; 500 nm for panel 3; and 250 nm for panel 4. **f** Heat map depicting transcription of genes encoding nuclear receptors (NR) and their co-regulators, using PCR arrays in B16F10 and SKMEL-28 cells treated or not with 2.5 μM DDA for 5 h, using PCR arrays. **g** Immunoblot for Nur77 and NOR1 protein expression in melanoma cells treated or not with DDA for 24 h. **h** Analysis of DDA cytotoxicity in B16F10 and SKMEL-28 cells transfected with control scramble siRNA (siSC), siNur77 and/or siNOR1. 72 h after transfection, cells were treated or not for 24 h with 2.5 μM DDA. Cell death is expressed as a percentage relative to the level of cell death induced by DDA in cells transfected with a scramble control siRNA (siSC). Data from **b–d** and **h** are the means ± S.E.M. of three independent experiments performed in triplicate, **P < 0.01, ***P < 0.001, t-test. All images and densitometry values are representative of three independent experiments

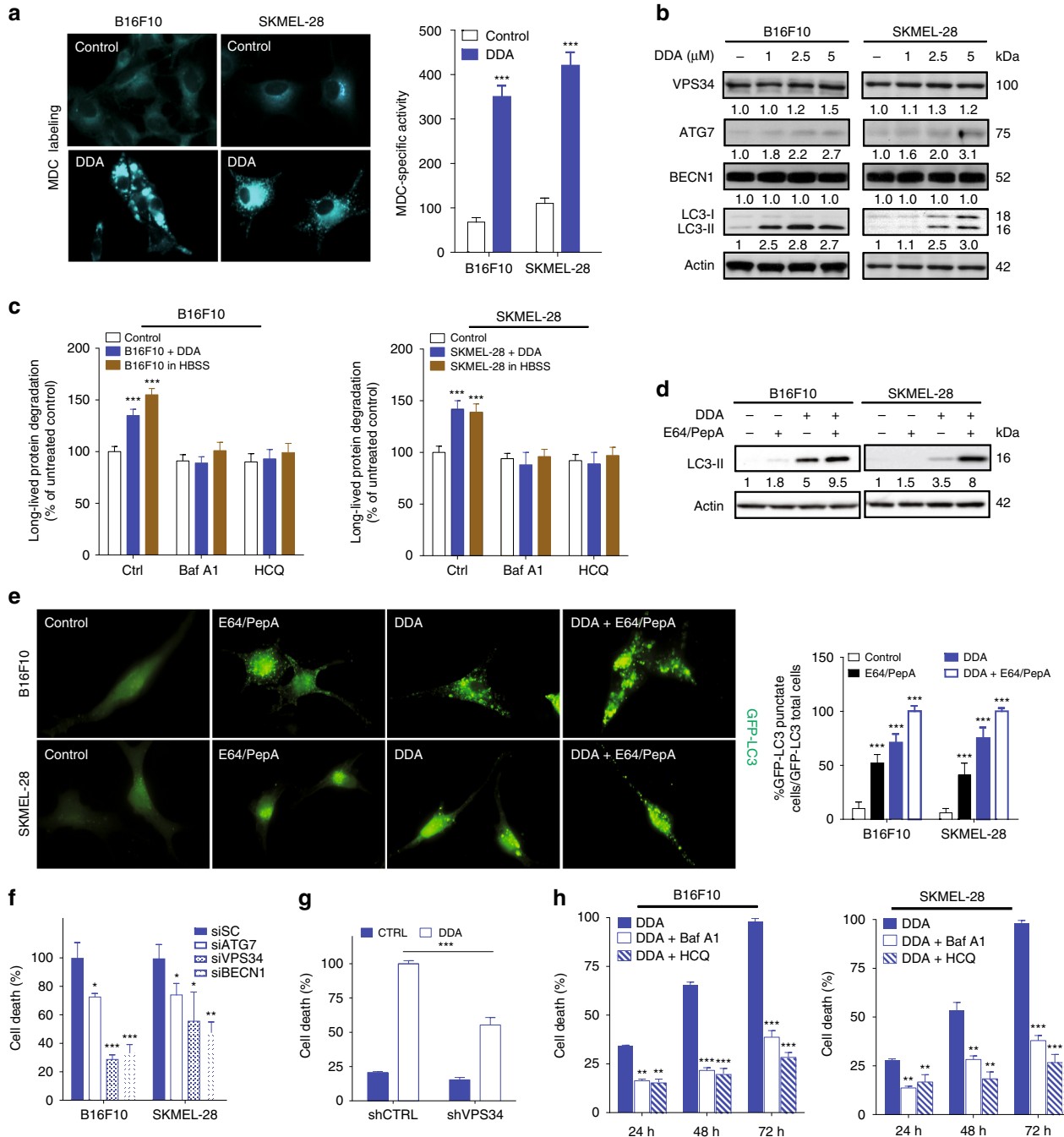

**Fig. 2** DDA induces Nur77- and NOR1-dependent lethal autophagy in melanoma cells. **a** DDA triggers the accumulation of autophagic vesicles. Cells were treated for 24 h with or without 2.5 μM DDA then stained with monodansylcadaverine (MDC) and observed by fluorescence microscopy. MDC-specific activity was measured by fluorescence photometry. **b** Cells were treated for 24 h with solvent vehicle or increasing concentrations of DDA then analyzed for autophagic protein expression by immunoblotting. Blots are representative of three independent experiments. **c** Long-life protein degradation was determined in cells treated with solvent vehicle (control) or 1 μM DDA for 18 h in the presence or absence of the autolysosomal inhibitors bafilomycin A1 (Baf A1) and hydroxychloroquine (HCQ). Autophagic activity was measured as the level of degradation of long-lived proteins. Starvation for 18 h in Hank's balanced salt solution (HBSS) was used as a positive control. **d** Immunoblots of LC3 proteins from cells treated for 24 h with or without 2.5 μM DDA and with or without E64 + pepstatin A (10 μg/ml). Images are representative of three independent experiments. **e** DDA induced the formation of punctate LC3 cells. Cells were transfected with a plasmid-expressing GFP-LC3 and then treated for 24 h with the solvent vehicle or 2.5 μM DDA, with or without E64 + pepstatin A (10 μg/ml) and observed by fluorescent microscopy. The percentage of GFP-LC3-positive cells with GFP-LC3 puncta was calculated. **f** Analysis of DDA cytotoxicity in cells transfected with scramble siRNA (siSC), siATG7, siVPS34, or siBECN1. Seventy-two hours after transfection, cells were treated or not for 24 h with 2.5 μM DDA. **g** Analysis of DDA cytotoxicity in SKMEL-28 cells permanently transfected with control shRNA (shCTRL) or shRNA against VPS34 (shVPS34). Cells were treated for 72 h with solvent vehicle (CTRL) or 2.5 μM DDA. **h** Cells were treated with 2.5 μM DDA for 24, 48 and 72 h in the presence or absence of the autolysosome inhibitors Baf A1 or HCQ. Cell death is expressed as in Fig. 1a. Data from **a**, **c**, **e**, **f**, **g** are the means ± S. E.M. of three experiments performed in triplicate, *$P < 0.05$, **$P < 0.01$, ***$P < 0.001$, t-test

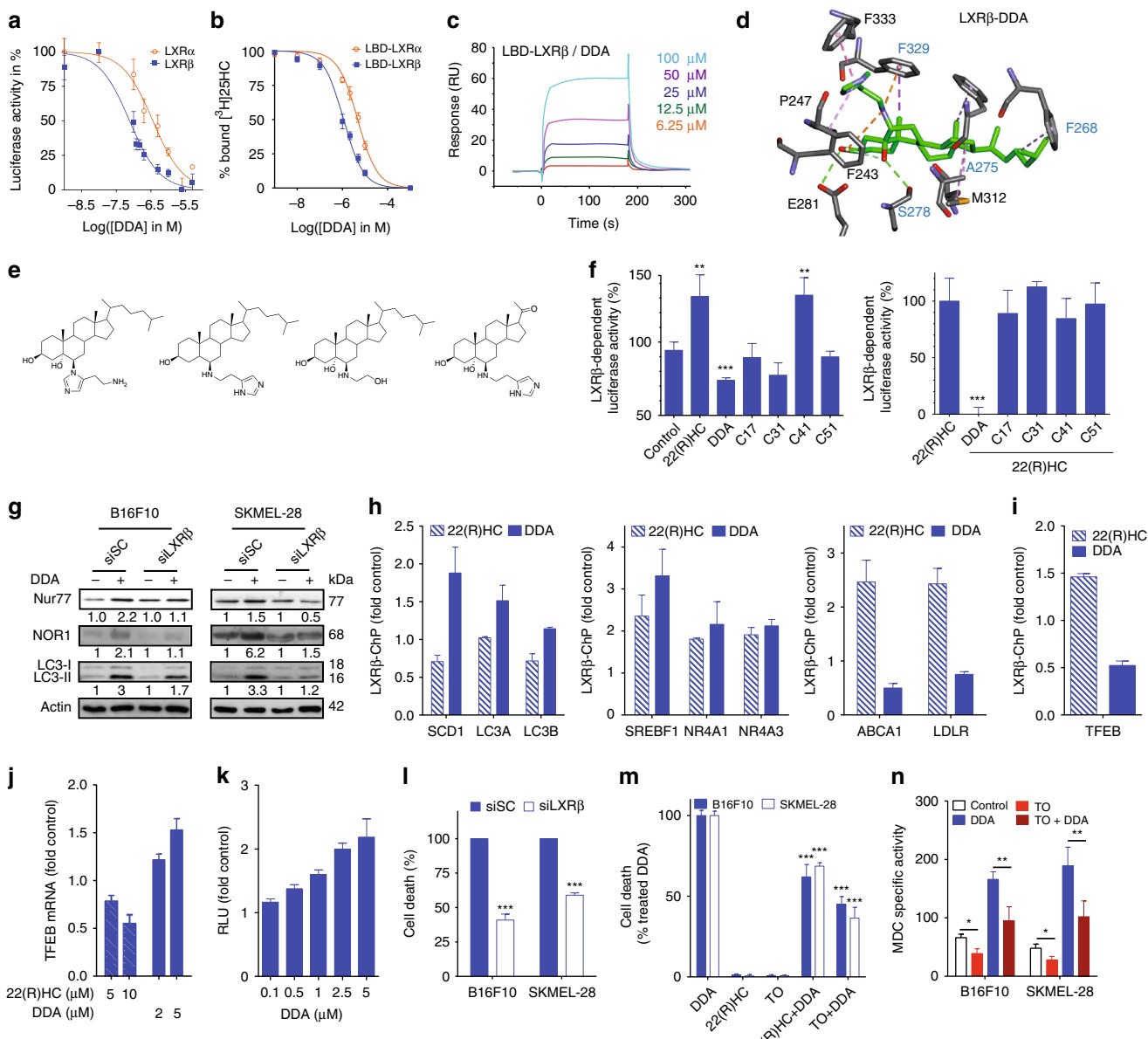

**Fig. 3** LXR are targets of DDA and LXRβ is required for its cytotoxicity in melanoma cells. **a** LXR transcriptional activity was analyzed using transient transfection reporter assays. Tranfected cells were treated with 10 μM 22(R)HC with or without DDA. **b** Competition binding assays on LBD-LXRα or LBD-LXRβ. **c** SPR sensorgrams showing the binding of DDA to the LBD-LXRβ. **d** Molecular docking of DDA with the LBD-LXRβ. Amino acid side chains that interact with DDA are represented (in black). The names of the amino acids known to interact with known LXR ligands are colored in blue. Gray: carbon atoms, white: hydrogen atoms, red: oxygen atoms, blue: nitrogen atoms, yellow: sulfur atoms. **e** Structures of DDA analogs assayed in the LXR reporter assay. **f** Analysis of LXRβ-dependent agonistic or antagonistic activities by DDA and analogs. **g** the stimulation of LC3, Nur77, and NOR1 protein expression by DDA is LXRβ-dependent. **h** ChIP-qPCR of LXRβ on the SCD1, LC3A, LC3B, SREBP1, NR4A1, NR4A3, ABCA1, and LDLR enhancers on SKMEL-28 cells treated or not with 10 μM 22(R)HC or 2.5 μM DDA. **i** ChIP-qPCR of LXRβ on the TFEB enhancer on SKMEL-28 cells treated or not with 10 μM 22(R)HC or 2.5 μM DDA. **j** Real-time PCR of TFEB expression in SKMEL-28 cells treated or not with 5 or 10 μM 22(R)HC, and 2 or 5 μM DDA. **k** Luciferase reporter gene assays with the TFEB promoter-luciferase construct in HEK293T. Cells were treated with increasing DDA concentrations. **l** Analysis of DDA cytotoxicity in cells transfected with control siRNA (shSC) or siLXRβ. Cells were treated with 2.5 μM DDA. **m** Analysis of the cytotoxicity of cells treated with or without 2 μM DDA, 5 μM 22(R)HC, 0.5 μM TO, 1 μM GW or 2 μM DDA + 10 μM 22(R)HC, 0.5 μM TO, or 1 μM GW. **n** TO reversed DDA induction of autophagic vesicles. Cells were treated for 24 h with or without 2 μM DDA, 0.5 μM TO, or 0.5 μM TO + 2 μM DDA. Cells were stained with MDC and observed by fluorescence microscopy. Data from **a**, **b**, **f**, **j**, **k**, **l**, **m** are the means ± S.E.M. of three independent experiments performed in triplicate (*P < 0.05, **P < 0.01, ***P < 0.001, t-test)

autophagy[13]. DDA-induced LC3-II formation was also inhibited by the single or double KD of NOR1 and Nur77 (Supplementary Fig. 3c–e). These data combined indicate that DDA induces lethal autophagy in melanoma cells and that this is mediated by Nur77 and NOR1.

**LXR is the molecular target of DDA.** To characterize the direct targets of DDA, we incubated B16F10 cells with [14C]-labeled DDA. After 6 h, DDA was located in the nucleus, suggesting its possible interaction with a nuclear receptor (NR) (Supplementary Fig. 3f). We therefore investigated the hypothesis that DDA could

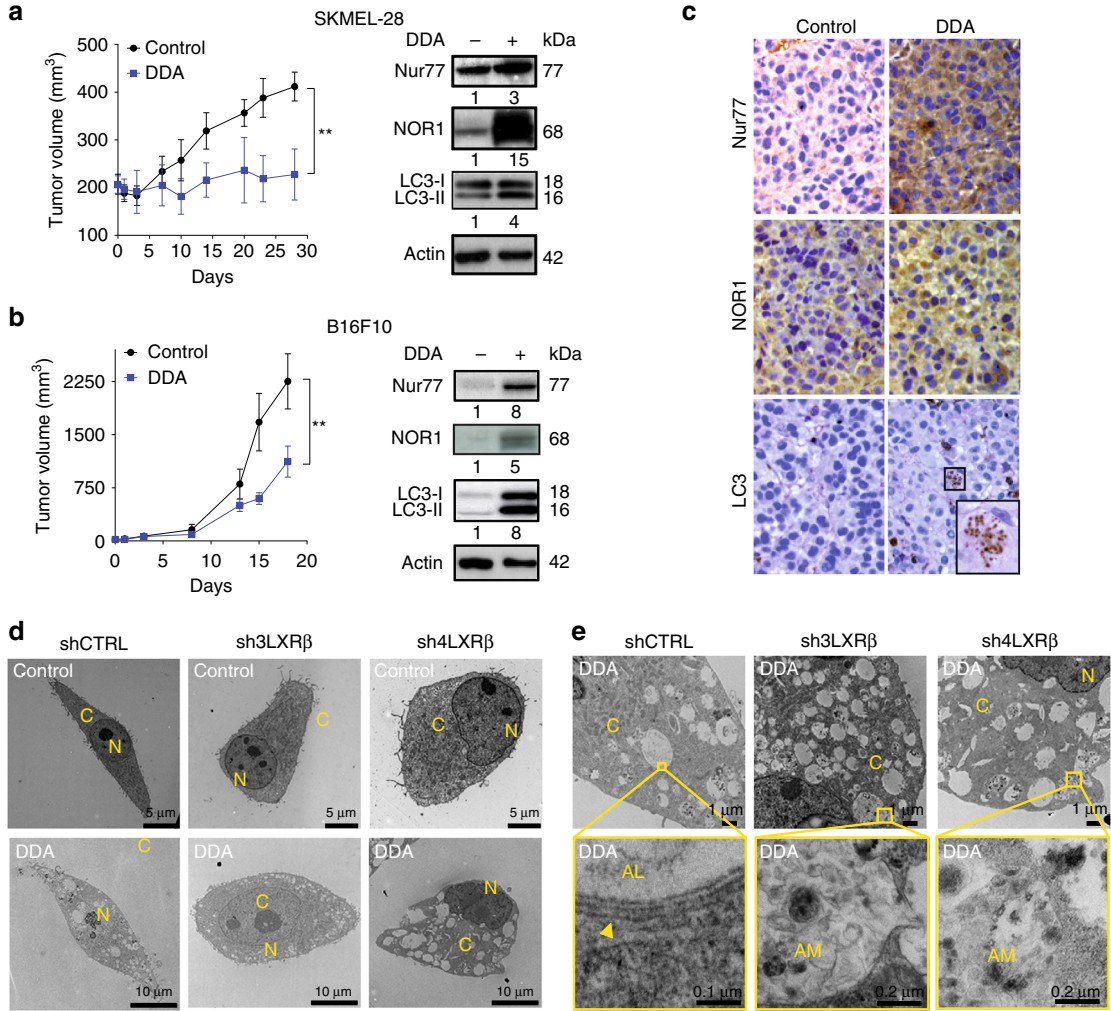

**Fig. 4** DDA induces autophagy in melanoma in vivo and in vitro in an LXRβ-dependent manner. **a**, **b** Mice engrafted with human SKMEL-28 or mouse B16F10 cells (10 per group) were treated with DDA (i.p. 20 mg/kg/day) or vehicle. Mean tumor volumes ± S.E.M. are shown, **$P < 0.01$, analysis of variance (ANOVA). Data are representative of three independent experiments. At the end of treatments, tumors were analyzed for Nur77, NOR1 and LC3 protein expression by **a**, **b** immunoblotting or **c** immunohistochemistry (brown staining). **c** Representative immunohistochemical analysis. Magnification ×40. Insert is 4× digital amplification. **d**, **e** TEM images of cells transfected with shCTRL, sh3LXRβ, or sh4LXRβ and treated with control vehicle or 2.5 μM DDA for 24 h. **e** TEM images of cells transfected with shCTRL, sh3LXRβ, or sh4LXRβ and treated with 2.5 μM DDA for 24 h. **d**, **e** N, nucleus; C, cytoplasm; AM, amphisome

modulate the liver-X-receptors (LXRs) as DDA-induced events such as triacylglycerol synthesis[5, 9] and NOR1 expression are also known to be regulated by the LXRs[27]. We found that DDA inhibited LXR-dependent luciferase activity stimulated by 22(R)-hydroxycholesterol (22(R)HC) in the presence of LXRα or LXRβ in a concentration-dependent manner, with $IC_{50}$ values of 362 ± 52 and 76 ± 12 nM, respectively (Fig. 3a). DDA did not show any agonistic or antagonistic transcriptional activity on any of the other NRs tested, establishing its selectivity for the LXRs (Supplementary Fig. 3g). Binding competition assay (Fig. 3b) and surface plasmon resonance assays on recombinant LXRα and LXRβ ligand-binding domains (LBD) (Supplementary Fig. 3h; Fig. 3c) indicated that DDA is a ligand of both isoforms with a fourfold preference for LXRβ-LBD, whereas GW3965 displayed a similar affinity for both LXR-LBD (Supplementary Fig. 3i). Docking experiments using the X-ray structures of LXRα and LXRβ also showed that DDA can be well accommodated within the LBD of both receptors (Fig. 3d; Supplementary Fig. 3j), with the ring-B hydroxyl in position 5α of DDA making hydrogen bonds with T302 (LXRα) and S278 (LXRβ), the secondary amine

on 6β making a hydrogen bond with S264 (LXRα), and cation–pi interactions between DDA and F243 and F329 (LXRβ). The imidazole ring of DDA is involved in an electrostatic bond with E267 (LXRα) and a π–π interaction with F333 (LXRβ), and the aliphatic side chain of DDA can make van der Waals interactions with H421, V425, W443, F257 and L439 (LXRα), and F268, A275 and M312 (LXRβ). Among these residues, E267, H421 and W443 in LXRα have been previously identified as being important for transcriptional activation, and H421 and W443 for the anchoring of the known LXR agonists TO901317 (TO) and 22(R)HC, whereas T302 has been previously described to stabilize TO binding[28]. The F268, A275, and M312 residues in LXRβ have been previously shown to interact with the LXR ligands T0 and GW3965 (GW); F329 was also shown to interact with GW and S278 with T0[29]. Modification or elimination of the chemical groups involved in the interaction of DDA with the LXRs (Fig. 3e) drastically affected the transcriptional activity of DDA (Fig. 3f; Supplementary Fig. 3k) inducing a switch from antagonist to agonist activity when eliminating (C41) or moving (C17) the imidazole ring on LXRα. C41 but not C17 induced a similar

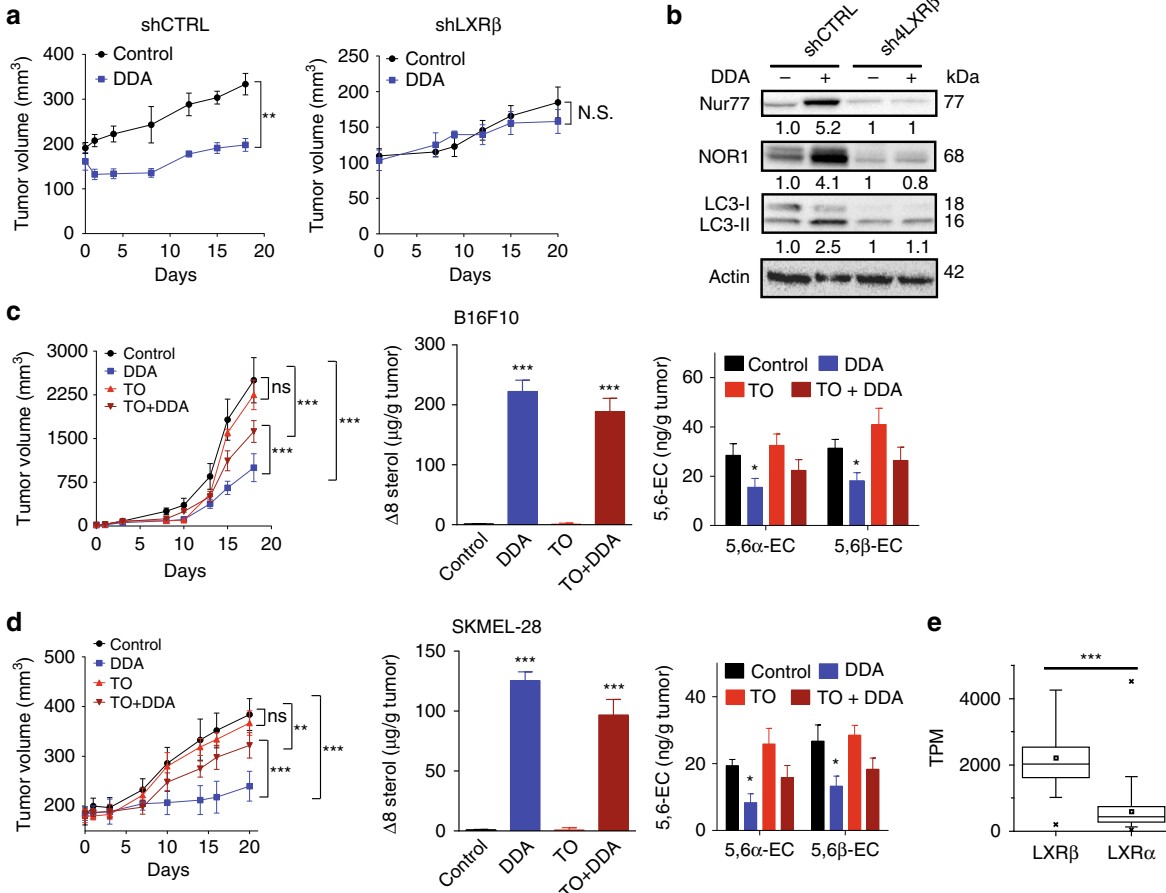

**Fig. 5** The anti-melanoma action of DDA in vivo is LXRβ-dependent. **a** Mice engrafted with SKMEL-28 cells transfected with shCTRL or sh4LXRβ (10 per group) were treated with DDA (i.p. 20 mg/kg/day) or vehicle. Mean tumor volumes ± S.E.M. are shown, **$P < 0.01$, analysis of variance (ANOVA). Data are representative of three independent experiments. **b** At the end of treatments, tumors were analyzed for Nur77, NOR1, and LC3 protein expression by immunoblotting. All images and blots are representative of three independent experiments. **c, d** Mice engrafted with mouse B16F10 cells or human SKMEL-28 (10 per group) were treated with vehicle, DDA (i.p. 20 mg/kg/day), TO (i.p. 20 mg/kg/day), and DDA + TO (i.p. 20 mg/kg/day each). Mean tumor volumes ± S.E.M. are shown, **$P < 0.01$, analysis of variance (ANOVA). Quantification of Δ8-sterols and 5,6α-EC and 5,6β-EC in tumors were quantified by GC/MS. The results are reported as μg Δ8-sterols or ng 5,6-EC/g tumors. **e** Box plot of TCGA RNA-seq data from patients with melanoma showing that LXRβ is the predominant LXR isoform expressed. ***$P < 0.001$

antagonist-agonist switch on LXRβ. This is consistent with a previous study showing that E267 was found to enhance transcriptional inhibition[28]. These data established that DDA is a non-selective endogenous ligand of LXRα and LXRβ isoforms. As shown in Supplementary Fig. 4a, DDA treatment increased the levels of mRNAs encoding ABCG5, LDLR, NOR1, Nur77, LC3A and LC3B. It also repressed ABCA1 mRNA levels and had no impact on SREBP1 mRNA levels. In comparison, the known LXR ligands 22(R)HC, GW, and T0, stimulated the transcription of most of the genes tested, but had no or a weaker effect than DDA on increasing NOR1, Nur77, LC3A, and LC3B levels. This was confirmed at the protein level for LC3-II (Supplementary Fig. 4b). KD of LXRβ, the only isoform expressed in these cells (Supplementary Fig. 4c), inhibited the stimulation of the expression of NOR1, Nur77 and LC3 at both the mRNA (Supplementary Fig. 4a) and protein (Fig. 3g) levels by DDA. In addition, the use of embryonic fibroblasts from LXRβ knock out mice confirmed the importance of LXR in DDA modulation of gene expression (Supplementary Fig. 4d). LXR agonists did not induce Δ8-sterols accumulation (Supplementary Fig. 4e) and KD of LXRβ in melanoma cells did not reversed DDA induction of Δ8-sterols accumulation (Supplementary Fig. 4f). We next assessed whether DDA affects the binding of LXRβ on the potent enhancers

identified in Supplementary Fig. 4g. DDA increased the binding of LXRβ on enhancers that are close to well-known target genes of LXRβ (SCD1, SREBF1), whereas it repressed LXRβ binding on other enhancers (ABCA1, LDLR), showing that DDA directly regulates LXRβ target genes (Fig. 3h). LXRβ binding on enhancers close to autophagic genes (LC3s and NR4As) is globally increased by DDA (Fig. 3h) strongly suggesting a direct control of their expression by DDA through LXRβ. 22(R)HC also regulates the binding of LXRβ on these enhancers reinforcing the possibility that LC3s and NR4As were LXRβ target genes. Importantly, differences appeared between DDA and 22(R)HC regarding the binding of LXRβ, in a gene-specific manner (Fig. 3h). This suggests that a set of LXRβ target genes is differently controlled by DDA compared to other LXR ligands and may explain why the anticancer mechanisms induced by DDA were never observed until now with LXR ligands. To gain insights into LXRβ target genes that might be involved in autophagy and lysosome biogenesis, we computationally predicted putative LXRβ-target genes based on the proximity between their Transcription Start Site (TSS) and LXRβ binding sites (LXRBS) (Supplementary Fig. 4g) according Wang et al.[30] methodology. We performed a gene ontology (GO) study using << Autophagy >> and << Lysosome organization >> terms on these putative LXRβ target genes and

got 26 hits including MAP1LC3B that could be regulated by LXR. Among them, TFEB was common to both series (Supplementary Fig. 4h). ChIP analyses of 22(R)HC and DDA-treated SKMEL-28

cells showed that they increased or inhibit, respectively, LXRβ binding to an enhancer located 5.7 kilobases from the TSS of TFEB (Fig. 3i) confirming that TFEB was a direct LXRβ-target

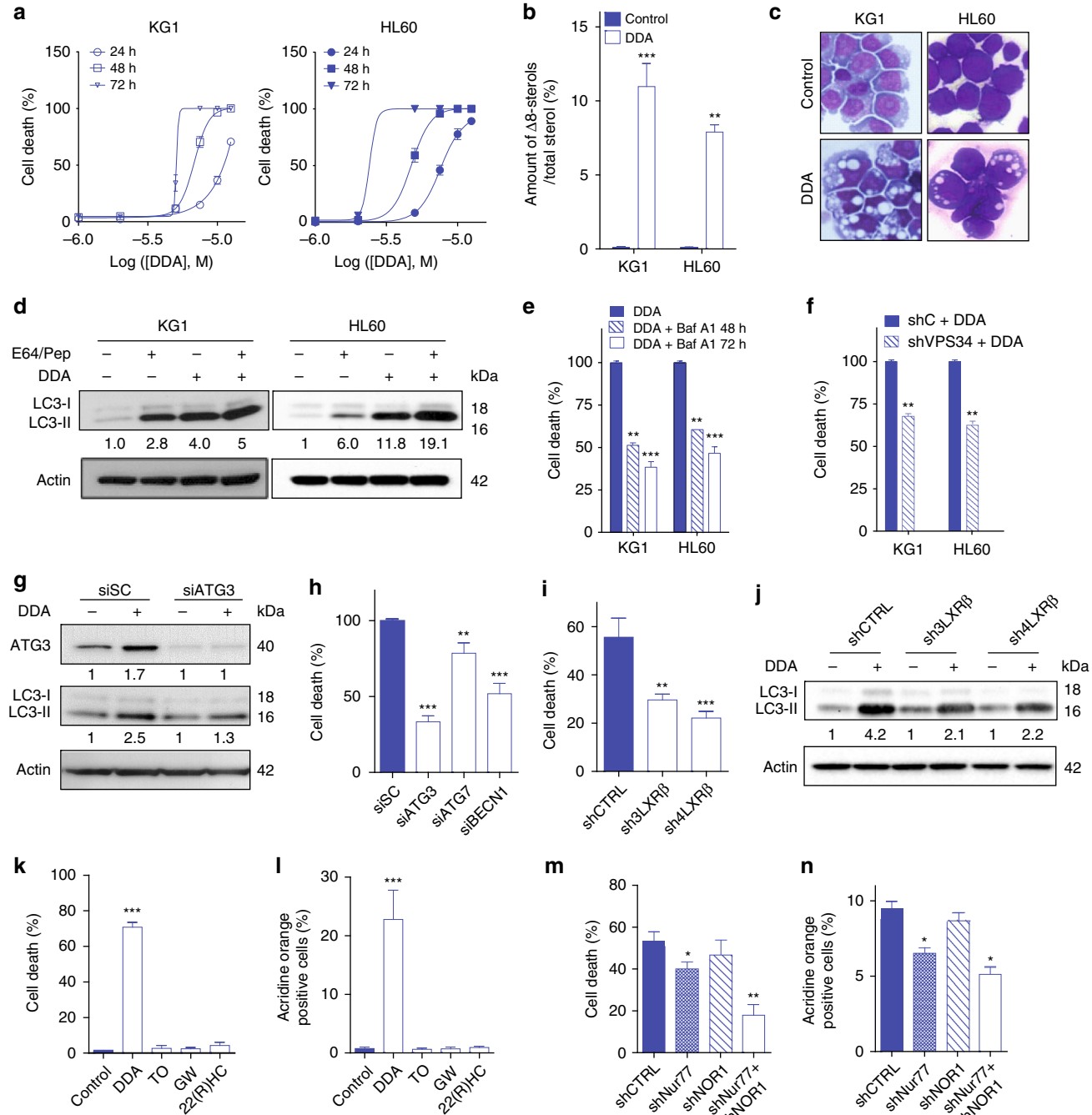

**Fig. 6** DDA induces lethal autophagy in AML cells via LXRβ. **a** DDA-induced cell death in KG1 and HL60 cells was determined over time as in Fig. 1b. **b** Quantification of the Δ8-sterols that had accumulated in KG1 and HL60 cells treated without or with 2.5 μM DDA. **c** May–Grünwald–Giemsa staining of KG1 and HL60 cells treated or not with DDA. **d** Immunoblots of LC3 proteins in cells treated or not with 2.5 μM DDA, and E64 + pepstatin A (Pep). **e** Effect of the pharmacological inhibitor of autophagy Baf A1 on DDA cytotoxicity at 48 and 72 h. **f** DDA cytotoxicity in KG1 or HL60 cells permanently transfected with control shRNA (shC) or shRNA against VPS34 (shVPS34) after 72 h treatment. **g** Immunoblots of ATG3 and LC3 proteins in KG1 cells transfected with control scramble siRNA (siSC) or siRNA against ATG3 (siATG3). Seventy-two hours after transfection, cells were treated for 24 h with 5 μM DDA. **h** Analysis of DDA cytotoxicity in KG1 cells transfected with control scramble siRNA (siSC), siATG3, siATG7, or siBECN1. Seventy-two hours after transfection, cells were treated for 24 h with 5 μM DDA. **i** Analysis of DDA cytotoxicity in KG1 cells transfected with shCTRL, sh3LXRβ, or sh4LXRβ. Cells were treated for 24 h with 5 μM DDA or vehicle. **j** Immunoblots of LC3 protein expression in cells transfected with shCTRL, sh3LXRβ, or sh4LXRβ and treated with 5 μM DDA or vehicle for 24 h. Analysis of the cytotoxicity **k** and acridine orange-positive vesicles **l** in KG1 cells treated or not with 5 μM DDA, 2 μM TO, 2 μM GW, or 10 μM 22(R)HC. The presence of Nur77 and NOR1 is required in DDA cytotoxicity (**m**) and autophagy (**n**). Data from **a**, **b**, **e**, **f**, **h**, **i**, **k**, **l**, **m**, **n** are the means ± S.E.M. of three experiments performed in triplicate, *$P < 0.05$, **$P < 0.01$, ***$P < 0.001$ t-test. All images and densitometry values are representative of three independent experiments

gene. DDA decreased LXRβ-binding to a TFEB enhancer and stimulates TFEB expression dose-dependently (Fig. 3j), 22(R)HC did the opposite, suggesting that LXRβ and LXRβ agonists act as repressors of TFEB and that DDA de-repressed this gene. Finally, the activation of TFEB expression by DDA increased the activity of a TFEB-dependent gene reporter assay (Fig. 3k) dose-

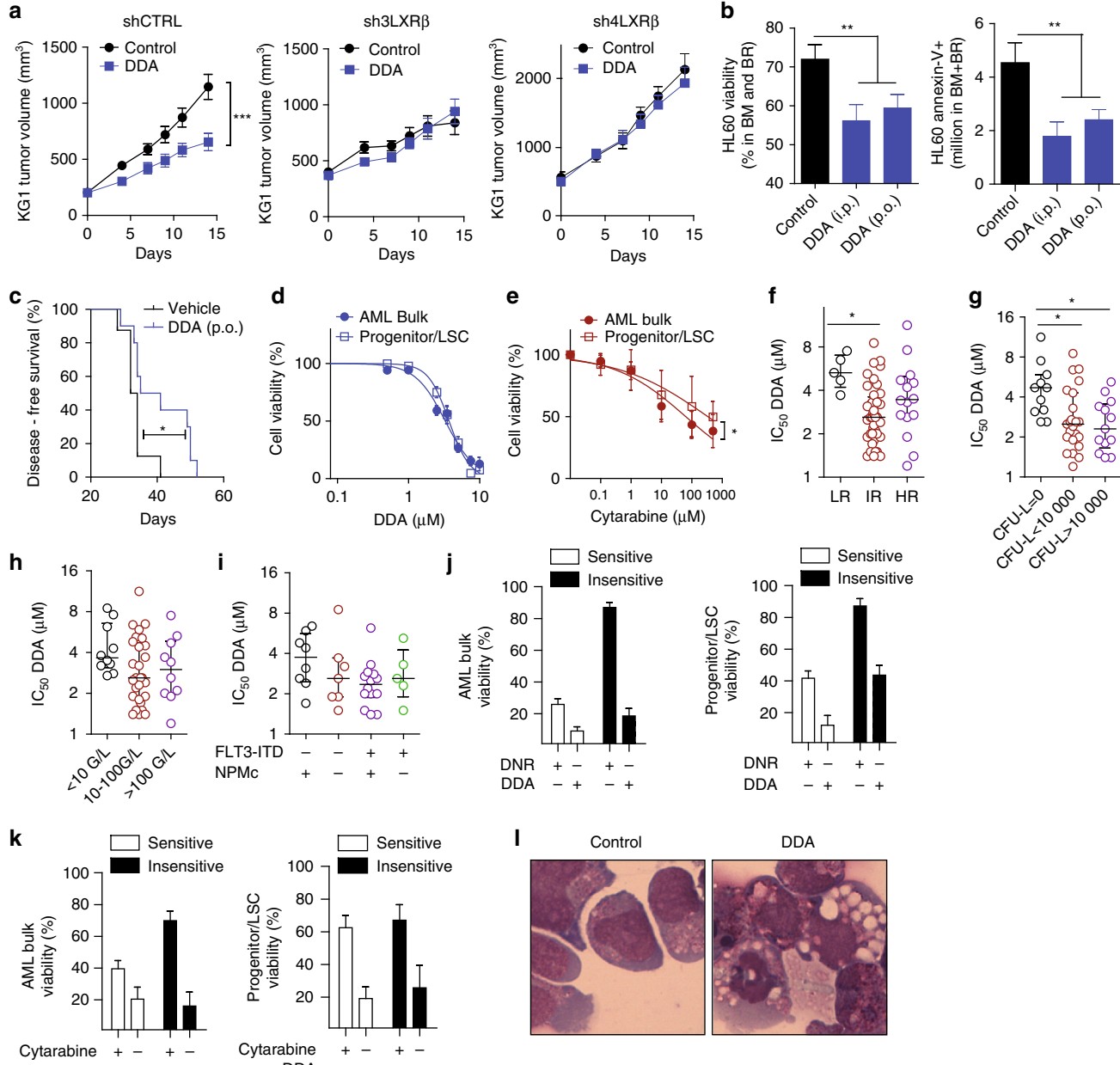

**Fig. 7** DDA induces an LXRβ-, Nur77-, and NOR1-dependent lethal autophagy in AML. **a** Tumor volume curves of xenografts of cells transfected with shCTRL, sh3LXRβ, and sh4LXRβ and implanted into NOD/SCID mice (20 per group) who were then treated daily with DDA (20 mg/kg/day, i.p.) or solvent vehicle. **b** HL60 cells were injected i.v. into irradiated NSG mice ($n = 10$ per group) who were then treated daily with DDA (20 mg/kg/day, i.p. or 40 mg/kg/day, p.o.) for 16 days. Analyses of HL60 cell contents and viability in bone marrow (BM) and brain (BR) of NSG mice. Cells were quantified by flow cytometry using human anti-CD45 and human anti-CD33 antibodies (left panel) and viability was determined by Annexin-V staining (right panel). **c** Overall survival was determined for NSG mice ($n = 10$ per group) engrafted with HL60 cells and treated, after disease establishment, with control (vehicle) or DDA (40 mg/kg/day, p.o.), *$P < 0.05$, log-rank test. Samples from AML patients ($n = 61$, Supplementary Data 1) were exposed to increasing concentrations of DDA (**d**) or cytarabine (**e**) for 48 h. Cell death was assessed both in the AML bulk and in the progenitor/LSC cells (CD34+CD38−CD123+) using Annexin-V/7AAD staining. Data are represented as the percentage of survival. Scatter plots comparing DDA efficacy in primary AML patients according to **f** their prognostic risk category (LR: low risk, IR: intermediate risk, and HR: high risk), **g** CFU-L formation, **h** total white blood count, and **i** their Flt3-ITD and NMP1 status. Samples from AML patients were exposed to 10 nM daunorubicin (DNR) **j** or 100 μM cytarabine **k** with or without 5 μM DDA for 72 h. "daunorubicin insensitive" or "cytarabine insensitive" when cell death was lower than 20%, and "sensitive" when cell death was over 70%. Bars represent S.E.M. **l** Images of primary AML cells stained with MGG after treatment with vehicle or DDA for 24 h. Images are representative of three independent experiments. Data from a-e are means ± S.E.M., and is representative of 3–5 independent experiments, **$P < 0.01$, t-test. **d–k** Bars represent S.E.M., *$P < 0.05$, t-test

dependently showing that DDA stimulated the transcriptional activity of TFEB. TFEB is a master transcription factor controlling genes involved in autophagy and lysosome organization and biogenesis[31–34]. These data revealed that LXR through DDA binding appears as a new player in autophagy and Ly biogenesis. Our results suggest that part of autophagy induction by DDA could be mediated through the de-repression of TFEB by the direct inhibition of LXRβ binding, in addition to the direct stimulation of the expression of NR4As genes by LXRβ.

We investigated the role of LXRβ in lethal autophagy mediated by DDA. In B16F10 and SKMEL-28 cells, KD of LXRβ expression significantly decreased DDA-induced cell death (Fig. 3l). In contrast, 22(R)HC, TO, and GW had no impact on cell death (Fig. 3m), consistent with work from Pencheva et al.[35] and partially reversed DDA cytotoxicity (Fig. 3m). As expected, we found that TO decreased the stimulation of autophagic vesicles formation by DDA (Fig. 3n). Altogether, these data establish that LXRβ controls DDA-induced lethal autophagy and NOR1, Nur77 and LC3 expression, and that DDA activity is distinct to that of the 22(R)HC, TO and GW LXR ligands.

**LXRβ is necessary for DDA anti-melanoma action in vivo.** DDA significantly reduced the growth of human and mouse melanoma tumors established in mice (Fig. 4a, b). Western blot analysis of tumor samples revealed an increased expression of Nur77, NOR1, and LC3-II in DDA-treated tumors compared to controls (Fig. 4a, b). This was confirmed by histochemical analysis of human SKMEL-28 tumors, where LC3-II appeared as brown puncta in the cytoplasm of cells from DDA-treated tumors (Fig. 4c). To determine whether LXRβ is involved in the antitumor activity of DDA in vivo, LXRβ was knocked down in SKMEL-28 cells using two different shRNA targeting LXRβ. Two clones (sh3LXRβ and sh4LXRβ), each with ~80% reduction in LXRβ mRNA and protein expression, were selected (Supplementary Fig. 4i) and both showed a lower capacity to induce LC3-II expression upon DDA treatment compared to control shRNA (shCTRL) (Supplementary Fig. 4j). The ultrastructure analysis of sh3LXRβ or sh4LXRβ cells treated with DDA revealed that DDA triggered the formation of Ly and amphisomes (AM) but not AL (Fig. 4d, e; Supplementary Fig. 5). This is in contrast to shCTRL-treated cells in which DDA induced AL, in addition to Ly and AM (Fig. 4d, e; Supplementary Fig. 5), and indicates that the DDA-mediated fusion between Ly and AM and between Ly and AP to form AL is LXRβ-dependent. In vivo, DDA significantly reduced the growth of established shCTRL tumors, but not that of shLXRβ tumors (Fig. 5a). Western blot analysis of tumor samples showed that DDA increased the levels of Nur77, NOR1, and LC3-II in shCTRL tumors and that these effects were decreased in shLXRβ tumors (Fig. 5b). We found that TO partially reversed the inhibition of tumor growth induced by DDA on melanoma cells (Fig. 5c, d). Analyses of the sterol profile in tumors confirmed the accumulation of Δ8-sterols in tumors, whereas no increase in 5,6-EC was measured (Fig. 5c, d) as observed on in vitro tests. LXRβ is the predominant isotype expressed in melanoma cells[35–38] and analyses of the TCGA dataset expression (Fig. 5e) corroborate these observations and supporting the importance of LXRβ targeting in melanoma cells. Altogether, these data demonstrate that LXRβ is essential to the induction of DDA-mediated lethal autophagy in vivo through the expression of LC3, Nur77, and NOR1.

**DDA triggers lethal autophagy in AML cells.** In mice, the abrogation of Nur77 and NOR1 has been shown to induce AML development[39]. The ability of DDA to reinstate the expression of these two factors in melanoma cells prompted us to assess the efficacy of DDA in inhibiting the progression of AML. In HL60 and KG1, two representative AML cell lines, DDA treatment triggered cell death (Fig. 6a), the accumulation of Δ8-sterols (Fig. 6b), the appearance of autophagic characteristics (Fig. 6c; Supplementary Fig. 6a–d), and autophagic flux (Fig. 6d). DDA also induced apoptotic characteristics (Supplementary Fig. 6e–i), however the caspase inhibitor z-VAD-fmk only weakly affected its cytotoxicity (Supplementary Fig. 6j). Both the pharmacological inhibition of autophagy and stable VPS34 KD significantly blocked DDA-induced cell death in long-term assays (Fig. 6e, f; Supplementary Fig. 6k). The KD of other autophagic proteins, ATG3, ATG7, BECN1, also inhibited cell death induced by DDA (Fig. 6g, h; supplementary Fig. 6l). LXRα and LXRβ were expressed in the two tested AML cell lines, but LXRα was weakly expressed compared to LXRβ (Supplementary Fig. 6m). Their KD (Supplementary Fig. 6n, o) showed that LXRβ mediated DDA-induced cell death (Fig. 6i; Supplementary Fig. 6n) and autophagy (Fig. 6j; Supplementary Fig. 6q, r) in these cells, which does not rule out a possible contribution of LXRα. LXR agonists were inefficient at inducing death and autophagy in AML cells (Fig. 6k, l; Supplementary Fig. 6s) and led to only a weak expression of Nur77 and NOR1 relative to DDA treatment (Supplementary Fig. 6t). In addition, Nur77 and NOR1 expression were induced by DDA in an LXRβ-dependent manner (Supplementary Fig. 7a, b), and both were required to control lethal autophagy (Fig. 6m, n; Supplementary Fig. 7c).

**DDA triggers an LXRβ-dependent lethal autophagy in AML.** DDA significantly reduced the growth and weight of KG1 or HL60 tumors implanted into NOD/SCID and nude mice (Supplementary Fig. 7d–f), and modulated the expression of autophagy markers LC3-II, P62 as well as Nur77 and NOR1 (Supplementary Fig. 7g). These effects were abrogated in LXRβ-KD KG1 cells but not in shCTRL cells (Fig. 7a; Supplementary Fig. 7h). DDA was found to accumulate in tumors (Supplementary Fig. 7f). The efficacy of DDA treatment (i.p. or p.o.) was evaluated in HL60 cells orthotopically engrafted (i.v.) into NOD/SCID/IL2rg (NSG) mice. Flow cytometry analysis using the human CD45 and CD33 markers revealed that DDA significantly reduced the number of HL60 cells (60.5 ± 11% for i.p. and 47 ± 9% for p.o.) and their viability (21.9 ± 6 % for i.p. and 17.3 ± 5 % for p.o.) in the bone marrow (BM) and brain (BR) (Fig. 7b). Importantly, DDA treatment significantly improved the survival of HL60-engrafted mice (Fig. 7c). To validate these results in AML patient tumors, we tested the effects of DDA on a panel of 61 AML patient samples (Supplementary Data 1). DDA significantly reduced cell viability (Fig. 7d) with a median $IC_{50}$ of 3.3 μM and a range of 1.2–11.3 μM. In contrast to the widely used AML drug cytarabine, DDA was as efficient at reducing viability in the progenitor/leukemic stem cell (LSC) (CD34+CD38–CD123+) subpopulation as in the AML bulk patient samples, and was 100-fold more efficient than cytarabine (Fig. 7e). Interestingly, the activity of DDA on AML patient samples did not correlate with the cytogenetic risk, clonogenic properties, white blood cell count or FLT3-ITD and NPM1 mutational status (Fig. 7f–i). In addition, AML patient samples were sensitive to DDA irrespective of their sensitivity to the two clinically used AML drugs daunorubicin (DNR) and cytarabine (Fig. 7j, k). LXRβ is the predominant isotype expressed in leukemia cell lines[38], which is corroborate by the analyses of the TCGA dataset (Supplementary Fig. 7j) supporting the importance of LXRβ targeting in AML cells. The treatment of primary AML patient samples with DDA revealed a massive vacuolation of cells (Fig. 7l) that was associated with increased LC3-II, Nur77 and NOR1 expression (Supplementary Fig. 7k–n).

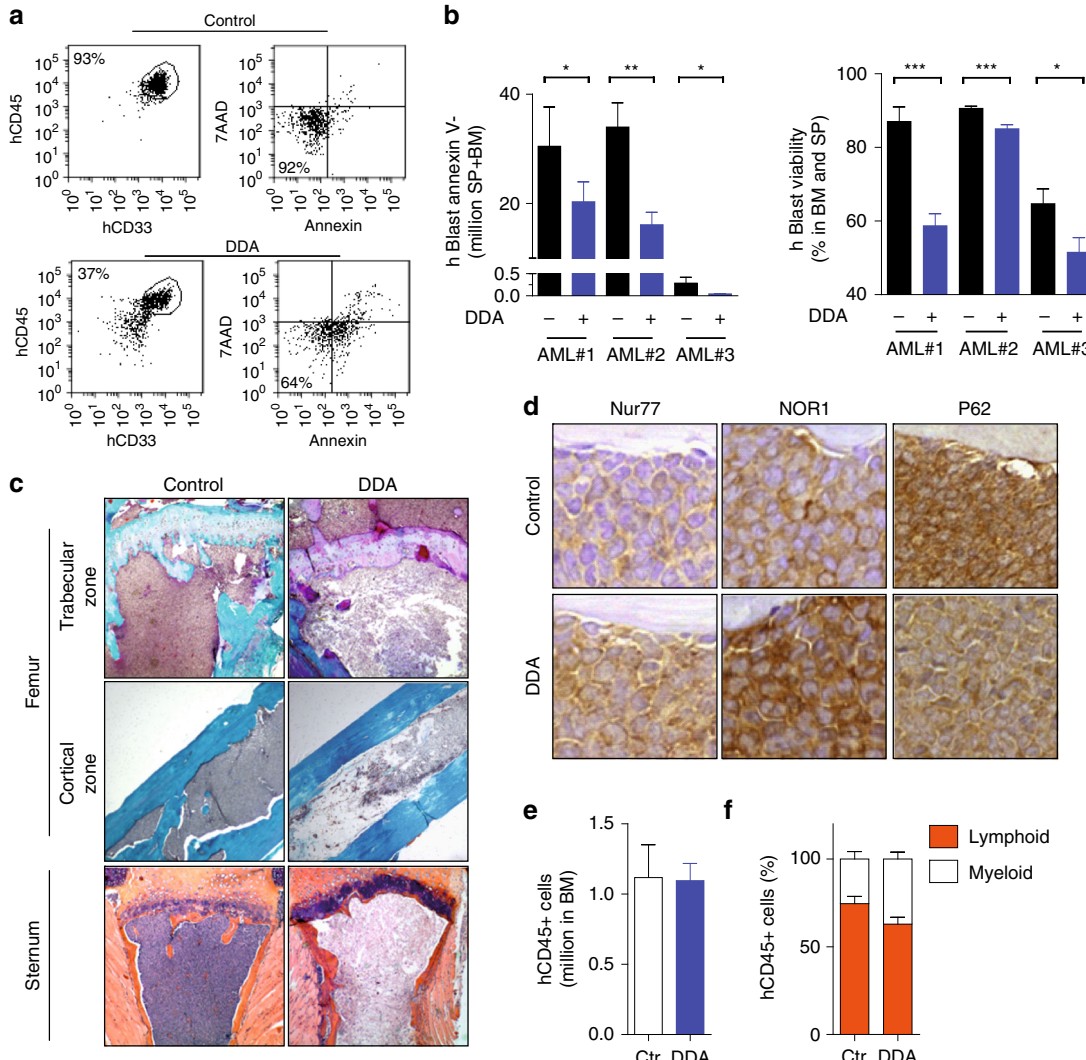

**Fig. 8** DDA exerts anti-leukemic activity in vivo in patient AML samples. Primary cells from AML patients were injected i.p. into irradiated NSG mice (three AML patients were tested separately). After validation of tumor engraftment, mice were treated with DDA (20 mg/kg/d, i.p.) or control (vehicle) for 19 days. **a** Representative flow cytometry analysis showing the selection of the human AML population (hCD45+hCD33+) and analysis of cell viability by Annexin-V/7AAD staining. **b** Human leukemic cell content in the hind limb bone marrow (BM) and spleen (SP) was measured by flow cytometry using human anti-CD45 and anti-CD33 antibodies. The viability of human CD45+CD33+ cells was determined by Annexin-V staining and flow cytometry analysis, $*P < 0.05$, $**P < 0.01$, $***P < 0.001$, $t$-test. **c** Histological analysis of femur and sternum sections from mice injected with primary AML cells (AML#1), stained with Goldner (femur) and HE (sternum). **d** Histological analysis of bone marrow staining for Nur77, NOR1, and P62. Normal human CD34+ cells from a healthy donor were implanted intravenously into NSG mice and treated with vehicle or DDA for 3 weeks (20 mg/kg/day, i.p.). **e** Engraftment was quantified by assessing the percentage of hCD45+ cells in the BM. **f** The percentage of human myeloid (CD45+/CD33+) and lymphoid (CD45+/CD19+) cells was determined by flow cytometry. **e**, **f** Bars represent S.E.M

These experiments suggest that the mechanism of action of DDA elucidated in AML cell lines and xenografts is conserved in AML patient samples.

**DDA is active on patient-derived AML xenografted in mice.** To determine the efficacy of DDA in a relevant model of established disease, primary AML cells from three patients were orthotopically engrafted (i.v.) into NSG mice. DDA significantly reduced the leukemic burden, the number of AML cells ($-57 \pm 15\%$) and their viability ($-19.6 \pm 7.6\%$) in both the bone marrow (BM) and spleen (SP), compared to controls (Fig. 8a, b). Histological analysis of the femur and sternum showed that control mice were completely packed with human primary AML cells in contrast to DDA-treated mice (Fig. 8c). BM recovered from DDA-treated mice showed high expression levels of Nur77 and NOR1 and

decreased levels of the marker of autophagy degradation, P62, compared to controls (Fig. 8d). Importantly, DDA had no effect on normal hematopoietic progenitor cells derived from human BM engrafted into NSG mice (Fig. 8e). The percentage of lymphoid (CD45$^+$ CD19$^+$) and myeloid (CD45$^+$ CD33$^+$) cells was not significantly different between DDA-treated and control mice (Fig. 8f). The pharmacokinetics of DDA was also determined in NSG mice after i.v., i.p. or p.o. administration. DDA plasma concentrations showed a flip-flop pharmacokinetic profile (Supplementary Fig. 8a, b), with elimination being far quicker than absorption ($t_{1/2}$ were 1.0 h vs 43 h for p.o and 0.9 h vs 19 h. for i.p. administration). The maximum sera concentrations ($C_{max}$) were 5.05 μM ± 0.87 μM for p.o and 6.10 ± 0.95 μM. for i.p. administration (Supplementary Fig. 8c). The calculated bioavailability of DDA, according to the area under curve methodology, was around 100% for p.o. and i.p. administration compared to i.v.

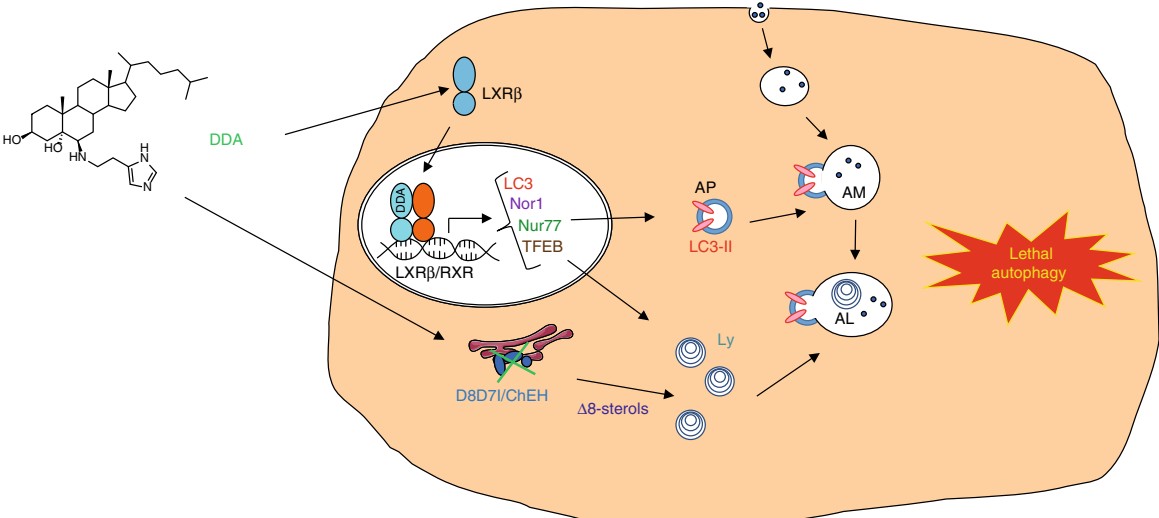

**Fig. 9** Molecular mechanisms through which DDA induced lethal autophagy in cancer cells. D8D7I, 3β-hydroxysterol-Δ8,7-isomerase; ChEH, cholesterol-5,6-epoxide hydrolase; Δ8-sterols, zymostenol and 8-DHC; Ly, lysosome; AM, amphisome; AL, autolysosome; AP, autophagosome

These concentrations correspond to active in vitro concentrations. After 21 days of treating NSG mice with DDA, DDA concentrations had risen to $1.32 \pm 0.03\,\mu M$ in the SP and $3.48 \pm 0.53\,\mu M$ in the BM (Supplementary Fig. 8d). These experiments demonstrate that in these experiments DDA was present at an active dose at the site of the origin and development of malignant myeloid progenitors, in accordance with its efficacy.

## Discussion

In the present study we provide evidence that DDA, a natural metabolite of cholesterol and histamine found in mammals, presents potent anti-tumor activities against melanoma and AML. These effects have been demonstrated both in vitro and in vivo, and include patient-derived xenografts, where DDA reduced tumor cell viability without causing toxic effects on the in vivo hematopoietic functions of normal human CD34+ progenitors. DDA was found to trigger lethal autophagy through an original mechanism by stimulating the expression of the pro-autophagic factors Nur77, NOR1, and LC3-II under the control of LXRβ. The screening of 61 different primary AML tumors from patients revealed that DDA is active in samples with high risk features including chemoresistance, molecular (FLT3-ITD) or cytogenetic (unfavorable karyotypes) lesions. Moreover, unlike cytarabine, DDA has a significant efficacy on the progenitor/LSC sub-compartment. In melanoma, the mechanism we have described seems to be independent of the constitutive Braf signaling pathway, thus our results indicate that DDA may offer an alternative route to improving therapies for these cancers.

We have established that DDA is a ligand of both LXR isoforms. In both melanoma and AML tumor models, we have shown that LXRβ is essential for mediating DDA-induced lethal autophagy. This is distinct from the three canonical LXR ligands tested, which did not stimulate lethal autophagy and did not or only weakly activated the transcription of the Nur77, NOR1, and LC3 genes. Moreover, the other LXR ligands tested did not inhibit D8D7I and caused zymostenol and 8-dehydrocholesterol accumulation, whereas the accumulation of these Δ8-sterols following DDA treatment contributed to the formation of Ly. Thus, the combined action of DDA on LXR and D8D7I most likely contributes to its high efficacy in triggering sustained lethal autophagy (Fig. 9), which is not observed with prototypical ChEH/D8D7I inhibitors or other LXR agonists.

Several studies have provided evidence that the LXRs are involved in the growth control of tumors and/or metastasis by acting on both the tumor cells themselves and their micro-environment[3, 36, 40–44], and LXR activation has been shown to have an effect on cell cycle arrest or apoptosis in different cancer cell lines and tumors[45, 46]. The inhibition of LXR-mediated glycolysis and lipogenesis by an inverse agonist ligand in colon cancer was also found to be associated with tumor growth inhibition and apoptosis[47]. Interestingly, synthetic LXR agonists such as GW inhibit primary melanoma progression by activating stromal LXRβ and causing ApoE secretion (a melanoma-extrinsic event), whereas their effects on metastasis are mediated through ApoE secretion in both the tumoral and stromal compartments[35]. We describe here that DDA controls melanoma and AML tumor progression through a distinct mechanism that has not been previously reported for a LXR ligand, and which involves tumor-intrinsic LXR activation to mediate lethal autophagy. In addition, our study revealed a yet unprecedentedly described relationship between LXR and the control of the expression of genes involved in autophagy and Ly biogenesis.

Autophagy is a well established survival mechanism that helps cells to escape to death and is also a mechanism of resistance to chemotherapy and endocrine therapies[22, 23, 48–50]. ChEH inhibitors Tam and PBPE stimulate a protective autophagy in melanoma cells and other cancer cells through D8D7I inhibition and Δ8-sterols accumulation[14, 15, 17, 23]. We report here that DDA mechanisms are different from these compounds. DDA inhibits D8D7I leading to Δ8-sterols accumulation and additionally binds to LXR and stimulate the expression of pro-autophagic genes. Thus, the activation of a multiple autophagic regulatory circuitry through new mechanisms with a single molecule may account for lethal autophagy as observed through combination therapy approaches[22, 51].

Advanced melanoma remains a fatal cancer with poor patient survival rates despite the development of immune checkpoint inhibitors and mutant Braf-targeted therapies, which despite having significantly improved relapse-free survival still encounter high levels of drug resistance and toxicity for immunotherapy[52, 53]. For decades, the therapeutic backbone of AML treatment has remained a combination of two genotoxic agents, namely daunorubicin and cytarabine, which induce fairly good remission rates but are unfortunately followed by frequent relapse, resulting in a poor outcome in most cases[54]. The prognosis for older AML

patients who are deemed unfit for chemotherapy is even worse. Thus, there is an urgent need to develop innovative therapies to improve the outcome of AML.

In conclusion, our results demonstrate the anti-melanoma and anti-AML efficacy of DDA that is independent of apoptosis and of the molecular and cytogenetic classifications of these cancers, and support the clinical evaluation of this new class of molecule. More generally, this study reveals the novel pharmacological control of lethal autophagy by a nuclear receptor through the action of a cholesterol metabolite.

## Methods

**Materials.** Chemicals and solvents were from Sigma unless otherwise specified. The caspase inhibitors z-VAD and z-DEVD-fmk were from Calbiochem. DDA and d6-DDA (DDA) were synthesized as previously described[5] and were confirmed as 99% pure by LC/MS. The compounds C17, C31, C41 and C51 were synthesized according to previously published procedures[5, 9, 55] and were confirmed as 99% pure by LC/MS. PBPE was synthesized as described previously[56].

**Cell culture.** HEK293T, B16F10 (Braf WT)[57], A375, WM35, WM-266-4, WM115, SKMEL-2, SKMEL-28 (Braf V600E mutated)[58], KG1 and HL60 cells were from the American Tissue Culture Collection and were cultured until passage 20–30. Cell lines were tested each month for mycoplasma contamination using MYCOALERT DETECTION Kit (Lonza, France). All cell lines were grown in a humidified atmosphere with 5% $CO_2$ at 37 °C in media containing 10% FBS (2% for WM35, 20% for KG1), penicillin and streptomycin (50 U/ml each) and 1.2 mM glutamine (3.2 mM for B16F10 and SKMEL-28). SKMEL-28 cells were grown in RPMI-1640 supplemented with heat-inactivated FBS. HEK293T, B16F10, A375, WM-266-4 and WM115 cells were grown in DMEM. WM35 cells were grown in a 4:1 mix of MCDB153:L15 with 5 µg/ml insulin and 1.68 µM $CaCl_2$. SKMEL-2 cells were grown in Eagle's Minimum Essential Medium. HL60 and KG1 cells and their transfectants were cultured in RPMI-1640 and IMDM medium respectively. Mouse embryonic fibroblasts (MEF) were derived from 13.5 dpc wild type or Lxrβ−/− embryos, as previously described[59]. Briefly, the dorsal part of the dissected embryos were sliced and incubated with a PBS trypsin-EDTA digestion mix (Sigma-Aldrich) at 37 °C for 45 min. These solutions were then homogenized using a syringe with a 19 G needle and plated onto DMEM (Sigma-Aldrich) supplemented with 10% FCS (Biowest), 2 mM glutamine (Sigma-Aldrich), MEM non-essential amino acid solution (Sigma-Aldrich), 100 µg/ml streptomycin (Sigma-Aldrich) and 100 µg/ml penicillin (Sigma-Aldrich). MEFs were repeatedly split upon confluency until they reached a period of massive senescence. Then, after 2 months of culture they regrew normally and were considered to be auto-immortalized.

**Animals.** Mice were handled and cared for according to the ethical guidelines of our institution and following the Guide for the Care and Use of Laboratory Animals (National Research Council, 1996) and the European Directive EEC/86/609, under the supervision of authorized investigators. All mice were maintained in specific pathogen-free conditions and were only included in protocols following 2 weeks of quarantine. NSG mice (NOD/LtSz-scid IL2Rγc null), NOD/SCID mice (NOD.CB17-Prkdcscid/), Nude mice (NU/NU) and C57Black6 mice were from Charles River Laboratories, Saint-Germain-sur-L'Arbresle, France.

**Assessment of apoptosis morphology.** After treatment with or without DDA for 24 h, cells grown on glass coverslips were washed once with ice-cold PBS then fixed with 1 ml of 4% paraformaldehyde for 20 min and washed once again with ice-cold PBS. The cells were then incubated with 1 ml PBS containing 300 nmol/l DAPI (Roche) for 10 min, washed twice with PBS and observed by fluorescence microscopy.

**Cell death assay.** Melanoma cells were seeded into 6-well plates at 60,000 cells per well and AML cells at 250,000 cells/ml. The cells were then treated with control solvent vehicle (1/1000 ethanol in water) or DDA. Where indicated, treated cells were incubated in the presence or absence of each inhibitor for time shown. Cell death was determined by the trypan blue exclusion assay. Cells were scraped and resuspended in trypan blue solution (0.25% (w/v) in PBS) and counted in a Malassez cell under light microscopy. Cell viability was then quantified by counting viable (trypan blue-negative) and dead cells (trypan blue-positive).

**Cytofluorometric analysis of melanoma cells.** To measure the ΔΨm, cells were incubated at 37 °C for 15 min in the presence of 40 nM DiOC6(3) (Molecular Probes) with or without 100 mM carbonyl cyanide m-chlorophenylhydrazone (CCCP). To determine superoxide anion generation, cells were kept at 37 °C for 15 min in the presence of 10 mM HE (Molecular Probes). After incubation with DiOC6(3) or HE, the cells were immediately analyzed by flow cytometry on a BD Facscalibur flow cytometer (BD Biosciences). DiOC6(3) was excited at 488 nm and

detected at 525 nm. HE was excited at 490 nm and detected at 620 nm. The median fluorescence intensity was quantified by FlowJo software.

**Cell cycle analysis.** Control and DDA-treated cells were fixed in 70% ethanol and then washed with PBS. Cells were incubated with 100 µg/ml RNase A for 30 min at 37 °C, permeabilized with 0.25% Tween-20, and stained with 50 µg/ml propidium iodide (PI) for 30 min at 37 °C. The DNA content of stained cells was analyzed by flow cytometry. The fraction of cells in sub-G1 phase was calculated using flowJo software (Ashland, Oregon). Data were obtained from $10^5$ viable cells.

**Annexin-V and PI staining.** Double staining for Annexin-V−FITC binding and DNA using PI was carried out in B16F10 and SKMEL-28 cells exposed to ethanol or DDA. Cells were washed in PBS and resuspended in the binding buffer (10 mM HEPES, 140 mM NaCl, 2.5 mM $CaCl_2$, 0.1% bovine serum albumin, pH 7.4). Cell suspensions were then incubated on ice with Annexin-V-FITC (Southern Biotech, Birmingham, AL, USA). After 15 min, an additional 380 µl of the binding buffer was added, followed by 0.5 mg/ml PI immediately before analysis with a BD Facscalibur flow cytometer (BD Biosciences). The percentage of Annexin-V−FITC/ PI-positive cells was determined using CellQuest software (BD Biosciences).

**NR and co-regulators PCR Arrays.** Human and mouse Nuclear Receptors & Co-regulators $RT^2$ Profiler™ PCR arrays were used to profile the expression of 84 genes encoding NR and their co-regulators (PAHS-056Y and PAMM-056Y, SABiosciences, USA), according to the manufacturer's instructions. COPS2 primers were replaced with NR4A3 primers in both arrays. qPCRs were run on an icycler iQ™ Real-Time Detection system (Bio-Rad). The relative abundance of each mRNA species was assessed following the manufacture's recommendations.

**Oxysterol analysis.** Oxysterol analysis in cell lines and tumors were done as previously described[12] with modifications[60].

**Microscopic autoradiography.** Cells were incubated with 1 µM [14C]-DDA for 6 h. Cells were washed with PBS and worked up as reported above. For TEM autoradiography the sections were covered with a thin carbon membrane and photographic emulsion (L4; Ilford, Limited Moberly Cheshire, England). After 12 weeks of exposure at 4 °C, the sections were developed (Kodak D19), fixed (30% Na-thiosulphate, pentahydrate, Merck 6516; Darmstadt, Germany), and evaluated by TEM (Philips, CM 100).

**Sterol analysis.** Sterols in cell homogenates were extracted with a solvent mixture containing chloroform/methanol 2/1 (v/v) spiked with epicoprostanol as the internal standard. Lipids were partitioned in chloroform after the addition of saline and saponified by methanolic potassium hydroxide (0.5 N, 60 °C, 15 min). The fatty acids released were methylated with BF3-methanol (12%, 60 °C, 15 min) to not interfere with the chromatography of sterols. The sterols were re-extracted in hexane and silylated, as described previously[61]. The trimethylsilylether derivatives of the sterols were separated by gas chromatography (GC) (Hewlett−Packard 6890 series) in a medium polarity capillary column RTX-65, (65% diphenyl 35% dimethyl polysiloxane, length 30 m, diameter 0.32 mm, film thickness 0.25 µm (Restesk, Evry, France)). The mass spectrometer (Agilent 5975 inert XL) in series with the GC was set up for the detection of positive ions. Ions were produced in the electron impact mode at 70 eV. Sterols were identified by the fragmentogram in the scanning mode and quantified by selective monitoring of the specific ions after normalization with the internal standard epicoprostanol and calibration with weighed standards.

**Filipin and Lamp1 staining.** Cells (75,000 per well) grown on glass coverslips were treated with 2.5 µM DDA for 48 h or with the solvent vehicle. Filipin staining was realized as described in[15]. For Lamp1 and Filipin staining, cells were fixed with 3.7% paraformaldehyde for 15 min at room temperature (RT), washed with PBS and permeabilized with 0.05% saponin in PBS with 1% BSA for 15 min. Cells were then washed with PBS and incubated for 1 h at RT with an anti-Lamp1 antibody (ab24170, Abcam, 1/500) (antibodies are listed on supplementary Table 4). After washing with PBS containing 0.05% saponin and 1% BSA, cells were incubated for 1 h at RT with Alexa fluor 488 conjugated-goat anti-rabbit antibodies (A-11008, Thermo Fisher Scientific). Cells were then washed with PBS containing 0.05% saponin and 1% BSA and incubated with Filipin (50 µg/ml, Sigma) for 75 min at RT before washing twice with PBS. Fluorescence images of cells were acquired with an LSM 780 (Zeiss) confocal microscope and a 63X Plan-Apochromat objectif (1.4 oil), equipped with a diode at 405 nm and an argon laser at 488 nm.

**Ultrastructural analysis.** Cells were fixed with 2% glutaraldehyde in 0.1 M Sorensen's phosphate buffer (pH 7.4) for 1 h and washed with the Sorensen's phosphate buffer (0.1 M) for 12 h. The cells were then post-fixed with 1% $OsO_4$ in Sorensen's phosphate buffer (Sorensen's phosphate 0.05 M, glucose 0.25 M, $OsO_4$ 1%) for 1 h, then washed twice with distilled water and pre-stained with an aqueous solution of 2% uranyl acetate for 12 h. Samples were then treated exactly as

described earlier[12]. Observations were performed with a Hitachi HT7700 transmission electron microscope. For quantification purposes 100 cells per grid were analyzed in triplicate for the presence of autophagic vesicles. Cells were considered positive when the number of vesicles exceeded 20 per cell.

**Cell staining**. Cells were treated with vehicle or DDA. The detection of autophagic vacuoles was carried out with MDC according to a previously published procedure[15]. Cells were incubated with 0.05 mM MDC for 60 min at 37 °C followed by fixation in 4% formaldehyde (15 min), and were then washed twice with PBS. The glass coverslips were mounted onto slides using Mowiol as a mounting medium. Analyses were carried out by fluorescent microscopy using a Zeiss LSM 510 microscope (Zeiss). For quantification purposes 100 cells per glass coverslip in triplicate were analyzed for the presence of fluorescent vesicles. Cells were considered to be positively stained when the number of fluorescent vesicles exceeded 20 per cell. For the morphological analysis of myeloid cells, cytospins were prepared by centrifugation in 150 μl PBS. Slides were stained at room temperature with May–Grünwald–Giemsa staining (MGG) and cellular morphology was examined using light microscopy. AML cells were stained with 10 ng/ml acridine orange for 15 min at 37 °C and subjected to flow cytometry analysis. In cells stained with acridine orange, the acidic compartments emit red fluorescence, the intensity of which is proportional to the degree of acidity. The cytoplasm and nucleoli emit green fluorescence.

**RNA isolation and analysis**. Total RNA was extracted from tumor cells or MEFs using an RNA Extraction kit (Qiagen) or Trizol (Invitrogen), respectively, and was quantified with NanoDrop. Complementary DNA (cDNA) synthesis was performed with 1 μg RNA iScript Reverse Transcriptase (Bio-Rad). 25 ng cDNA was amplified using SyBR Supermix (Bio-Rad). RNA expression of the different genes of interest (Supplementary Tables 1, 2) were determined using the ΔΔCT method, where the difference in the threshold cycle (ΔCT) values of the target gene and the housekeeping gene for each sample was normalized to the ΔCT value of the untreated sample.

**Immunoblots**. Immunoblotting was carried out as described previously[5]. Proteins were separated on SDS-PAGE gels, electro-transferred onto polyvinylidene difluoride membranes and incubated overnight at 4 °C with the following antibodies: anti-Atg3, anti-Atg5 (D1G9), anti-Atg7 (D12B11), anti-Beclin (D40C5), anti-caspase-3 (Cell Signaling), anti-Bcl-2 (Millipore), anti-Bax (Millipore), anti-LC3 (Sigma-Aldrich), anti-LXRα (SantaCruz), anti-LXRβ (SantaCruz), anti-Nur77 (Active motif), anti-NOR1 (R&D systems), anti-PARP (Cell Signaling), anti-Vps34 (Cell Signaling), or anti-Actin (Millipore). For visualization, an ECL plus kit (Amersham Biosciences) was used, and chemiluminescence was measured by autoradiography (Supplementary Fig. 9). Specific bands were quantified with ImageQuant software.

**GFP-LC3 staining**. B16F10 and SKMEL-28 cells were transfected with a plasmid expression vector encoding GFP-LC3 (generously provided by Dr P. Codogno), using the FuGENE 6 Transfection Reagent (Roche Applied Science). At the indicated times after transfection, GFP-LC3 staining was visualized using a Zeiss LSM 510 fluorescent microscope (Zeiss). For quantification purposes 100 cells per glass coverslip in triplicate were analyzed for the presence of GFP-LC3 puncta. Cells were considered to be positively stained when the number of GFP-LC3 puncta exceeded 20/cell. The ratio of the GFP-LC3 puncta positive cells per 100 cells was calculated.

**Analysis of protein degradation**. Long-lived protein degradation was measured as described previously[15, 62]. Cells were incubated for 18 h at 37 °C with 0.2 μCi/ml of [$^{14}$C]L-valine in complete medium. After 5 h incubation with fresh medium supplemented with 10 mM cold valine, fresh chase medium with or without 1 μM DDA (18 h or 24 h) Bafilomycin A1 (10 nM) or HCQ (40 μM) were added. Cells and radiolabeled proteins were precipitated with trichloroacetic acid at a final concentration of 10% (v/v) at 4 °C. Radioactivity was measured by liquid scintillation counting. The protein degradation of long-lived proteins was calculated from the ratio of the acid-soluble radioactivity in the medium to that in the acid-precipitable cell fraction.

**Gene reporter luciferase assays**. HEK293T were used for luciferase assays. Cells were transfected using the PolyEthylenImine (PEI) method with human retinoid-X receptor (RXRγ) and human liver-X-receptors (LXRα or LXRβ) with LXRE-Luc; RXRγ and human farnesoid X receptor (FXR) with EcRE-Luc; RXRγ and human steroid and xenobiotic X receptor (SXR) with DR4-Luc; RXRγ and human peroxisome proliferator activated receptors (PPARα or PPARγ) with PPRE-Luc; RXRγ and human retinoic acid receptor (RARα) with DR1-Luc; human AHR and human ARNT with a DRE-Luc; human vitamin D receptor (VDR) with VDRE-Luc. Cells were transfected with plasmids encoding human glucocorticoid receptor (GR), progesterone receptor (PR), or androgen receptor (AR) with MMTV-Luc; or estrogen receptors (ERα and ERβ) with an ERE-Luc. The day after transfection with the luciferase reporter gene construct, cells were seeded in 12-well plates

(50,000 cells per well). After 6 h, cells were then treated with the following ligands dissolved in ethanol: 10 nM estradiol (E2) for ERα and ERβ; 10 nM R1881 for AR; 10 nM R5020 for PR; 10 nM dexamethasone for GR; 10 μM fenofibrate for PPARα; 100 nM rosiglitazone (BRL 49653) for PPARγ; 100 nM cis-retinoic acid (cRA) for RAR; 10 nM dioxine for AHR, 10 μM rifampicine for SXR and 10 μM 22(R) hydroxycholesterol (22(R)HC) for LXRα and LXRβ. At the end of each treatment, cells were lysed in 100 μl lysis buffer (Promega). Luciferase activity was measured using the luciferase assay reagent (Promega), according to the manufacturer's instructions. Protein concentrations were measured using the Bradford method to normalize luciferase activities. For each condition, luciferase activity was calculated from the data of three independent wells.

**Binding assay**. For each point, 0.4 μCi [$^3$H]-25-hydroxycholesterol ([$^3$H]-25HC) (specific activity 80.0 Ci/mmol, PerkinElmer) and 250 ng GST-LBD-LXRα (Invitrogen) or 250 ng GST-LBD-LXRβ (Invitrogen) were used. Concentrations of DDA and 25HC ranging from 1 nM to 100 μM were diluted in binding buffer (Tris 10 mM pH 7.5, EDTA 1.5 mM, DTT 2 mM, CHAPS 2 mM, gamma-globulin 10 mg/ml). Recombinant LBDs were added to a mixture of DDA and [$^3$H]-25HC, gently mixed by pipetting, and then incubated for 24 h at 4 °C with 300 rpm agitation. Charcoal-coated dextran (5%) in binding buffer without gamma-globulin was added, gently mixed by pipetting, and then incubated for 15 min at 4 °C with 400 rpm agitation. Total specific activity was measured for each point and the samples were then centrifuged for 10 min at 10,000 rpm at 4 °C. Supernatants were collected and the specific activity of each supernatant was measured and corresponded to the amount of bound [$^3$H]-25HC. This value was normalized using the total specific activity measured for each point. Non-specific binding was determined by measuring the amount of bound [$^3$H]-25HC in the presence of a saturating concentration of cold 25HC. The results are expressed as the percentage of [$^3$H]-25HC bound to LBDs relative to its binding without a competitive ligand.

**Surface plasmon resonance assays**. All binding studies based on SPR technology were performed on BIAcore T200 optical biosensor instrument (GE Healthcare). Immobilization of the GST-tagged ligand-binding domains of human LRXβ (LBD-LRXβ, PV4660, Thermo Fisher Scientific, MA, USA), human LRXα (LRXα-LBD, PV4657, Thermo Fisher Scientific, MA, USA) and GR (GR-LBD, A15668, Thermo Fisher Scientific, MA, USA) was performed by the covalent coupling of the ligand to the chip surface using an amine coupling (CM5) sensorchip in PBS-P + buffer (20 mM Phosphate Buffer pH 7.4, 2.7 mM KCl, 137 mM NaCl, and 0.05% surfactant P20) (GE Healthcare). All immobilization steps were performed at a flow rate of 10 μl/min with a final concentration of 20 μg/ml. The total amount of immobilized protein was 11,000–12,000RU. Immobilization of GR-LBD was performed on channel Fc1 that was used as a reference surface for non-specific binding measurements. A Low Mass Weight-Multiple-cycle kinetics (LMW-MCK) analysis to determine affinity constants (Kd) was carried out by injecting different protein concentrations (6.25–500 μM). Binding parameters were obtained by fitting the overlaid sensorgrams with the 1:1 Langmuir binding model of the Biacore T200 Evaluation software version 2.0 or Steady State affinity model of the BIAevaluation software version 3.1.

**Molecular modeling**. All molecular modeling studies were conducted using Accelrys Discovery Studio 4.0 (Accelrys Software, Inc., San Diego, CA; http://accelrys.com). Crystal structure coordinates were obtained from the protein data bank (http://www.pdb.org). DDA was prepositioned in the TO091317-LXRα LBD (PDB code: 1UHL)[28] and in the TO091317-LXRβ LBD crystal structures (PDB code: 1PQ9)[29]. Once prepositioned, TO091317 was unmerged from both complexes and deleted, whereas DDA was merged to the LBDs. The resulting complexes were submitted to energy minimization using 250 steps of the steepest descent followed by a conjugated gradient until the root mean square gradient was less than 0.001 kcal/mol/Å. A distant-dependent dielectric term ($\epsilon = r$) and 20-Å nonbonded cut-off distance were chosen, whereas the hydrogen bond involved in the conformation of the α helices was preserved by applying a generic distance constraint between the backbone oxygen atoms of residue $i$ and the backbone nitrogen atoms of residue $i + 4$, excluding prolines. The minimized coordinates of the receptor were then used as the starting point for 100 ps at 300k using the Verlet algorithm whereas the constraint used during minimization was maintained. The resulting conformation was then further minimized using 250 steps of the steepest descent followed by a conjugated gradient until the root mean square gradient was less than 0.001 kcal/mol/Å, as was previously done for estrogen receptors[63, 64].

**Chromatin immunoprecipitation**. Experiments were performed as reported previously[65]. Cells plated in 15 cm-dishes were cross-linked with 1% formaldehyde in medium for 10 min under gentle rotation. The crosslinking reaction was stopped by the addition of 125 mM L-glycine and the cells were washed thrice with ice-cold PBS. They were harvested in lysis buffer 1 (50 mM Hepes-KOH pH 7.5, 140 mM NaCl, 1 mM EDTA, 10% glycerol, 0.5% of NP-40 and 0.25% Triton X-100), incubated on a rotating wheel for 10 min at 4 °C and nuclei were pelleted at 2000×$g$ for 5 min. Pellets of nuclei were resuspended in lysis buffer 2 (10 mM Tris-HCl pH 8.0, 200 mM NaCl, 1 mM EDTA and 0.5 mM EGTA), incubated on a rotating

wheel for 5 min and pelleted at 2000 g for 5 min. Pellets were lysed with lysis buffer 3 (10 mM Tris-HCl pH 8.0, 100 mM NaCl, 1 mM EDTA, 0.5 mM EGTA, 0.1% Na-deoxycholate, 0.5% N-lauroylsarcosine and a protease inhibitor cocktail (Roche)) and chromatin was sheared by 30 cycles of 30 s of sonication at high power with a bioruptor sonicator. 0.1% Triton X-100 was added, cell membranes were discarded by centrifugation at 16,000×g for 10 min and nuclear extracts were collected. 10 μl of nuclear extracts were saved to measure the inputs and the rest was incubated overnight at 4 °C on a rotating wheel with dynabeads that were pre-coated during 3 h with a LXRβ antibody (Active Motif) in PBS containing 0.5% BSA. Dynabeads were harvested with a magnetic stand and washed 10 times with RIPA buffer (50 mM Hepes-KOH pH 7.5, 500 mM LiCl, 1 mM EDTA, 1% of NP-40, 0.7% Na-deoxycolate). A last wash with TBS was done and reverse-crosslinking of the inputs and of the dynabeads were performed with the elution buffer (50 mM Tris-HCl pH 8, 10 mM EDTA, and 1% SDS) at 65 °C overnight under agitation at 900 rpm. Eluates were separated from the dynabeads with a magnetic stand and collected in new tubes. Samples were diluted in TE buffer and incubated with 25 μg/ml RNase for 1 h at 37 °C followed by an incubation with 200 μg/ml proteinase K for 2 h at 55 °C. DNA was isolated with phenol:chloroform:isoamyl alcohol (25:24:1), transferred to Phase Lock Gel Light tubes (5 PRIME) and centrifuged at 10,000 rpm for 5 min. Top phases were collected in new tubes containing 10 μg glycogen in 200 mM NaCl. Absolute ethanol was added, samples were mixed, incubated 20 min at −80 °C and centrifuged at 16,000×g for 30 min at 4 °C. Pellets were washed with 80% ethanol, centrifuged at 16,000×g for 5 min and dried 30 min at room temperature. Pellets were resuspended in nuclease-free water. Quantitative PCR was performed with specific primers (Supplementary Table 3). ChIP values were standardized with the GAPDH coding region as an internal control and normalized with the input values.

**Small interfering RNA transfection**. The expression of endogenous genes of interest was silenced with a pool of four siRNAs (Dharmacon) along with a control scrambled sequence siRNA (Dharmacon) in melanoma or AML cells, as indicated. B16F10 and SKMEL-28 cells were seeded in 100 mm-dishes (300,000 cells/dish for B16F10 cells, 500,000 cells/dish for SKMEL-28 cells). Twenty four hours after seeding cells were transfected in OptiMEM with 50 nmol/l of control siRNA or specific siRNAs using DharmaFECT1 (Dharmacon) and following the procedure recommended by the manufacturer. KG1 cells were transfected with 70 nM/$10^6$ cells of the indicated siRNA with the NEON Transfection System (Invitrogen), according to the supplier's instructions.

**Stable ShRNA transfection**. For the KD of all the genes (except Nur77 and NOR1 in KG1 cells), KG1 or SKMEL-28 cells were transfected with three different puromycin-resistant shRNA plasmid specific to the gene of interest or with a non-specific shRNA plasmid (SureSilencing shRNA, Qiagen, Courtaboeuf, France), using the NEON transfection system (Invitrogen) and according to the supplier's instructions. Selection of the stable clones was obtained after growing the cells for 1 month in the presence of puromycin (0.5 μg/ml). For the KD of Nur77 and NOR1 in KG1, 29 mer shRNA constructs in pRS retroviral vectors targeting Nur77 and NOR1 were purchased from Origene. HEK293T cells co-transfected with viral protein encoding plasmids containing Nur77, NOR1 or control genes were used. Supernatants containing retroviruses were collected. KG1 cells were plated in serum-free medium with the retroviral supernatant of interest. After 4 h of incubation, cells were spinoculated (2300 rpm, 90 min) and resuspended in 20% FBS medium. Pools of stables clones were obtained after growing the cells for 3 weeks in the presence of puromycin (1 μg/ml).

**Cell transfection**. One day after seeding in 100 mm-dishes (300,000 cells/dish), B16F10 cells were transfected with a DR4-Luciferase reporter plasmid, an RXRα-encoding plasmid (PSG5-RXR) and an LXRα-encoding plasmid or an LXRβ-coding plasmid by using the PolyEthylenImine (PEI) method[26]. For each dish, a transfection solution was prepared with 4.3 μg of PEI and 5 μg of plasmid in 3 ml of OptiMEM and incubated with cells for 5 h at 37 °C. Medium was then replaced by complete medium and cells were incubated at 37 °C in a humidified 5% $CO_2$ incubator.

**ChIP-seq data re-analyses**. The dataset GEO: GSE77039[66], available publicly on the NCBI server, was used. SRA files from the dataset were downloaded from the NCBI server, converted into fastq files and mapped to the human reference genome hg19 with bwa mem. Visualization of ChIP-seq profiles on defined regions of the genome was performed with the Integrative Genomics Viewer (version 2.3.67).

The dataset GEO: GSE77039 (Savic et al.[66]) was processed and analyzed as described previously (Segala et al.[65]). Genes located less than ten kilobases from the LXRβ binding site were identified as potent LXRβ target genes with the BETA-minus tool from the Cistrome framework. The study of gene ontology was performed with HumanMine v3.2.

**Annexin-V/7AAD assay**. The cytotoxic response to DDA of AML cells was determined by flow cytometry. Cells were exposed to DDA for 24 or 48 h then washed with cold phosphate-buffered saline (PBS) and resuspended in 200 μl of Annexin-V binding buffer. For the analysis of AML patient samples, cells were

incubated with antibodies against the surface markers CD45-V450 (clone H130), CD34-PeCy7 (clone 8G12), CD38-APC (clone HB7) and CD123-PE (clone 9F5) (BD Pharmagen) before being resuspended in Annexin-V binding buffer. Annexin-V-fluorescein isothiocyanate (BD Pharmagen, clone 2331) and 7-aminoactinomycin D (7AAD) were then added, and the samples were incubated in the dark at room temperature for 15 min. The percentage of viable cells, Annexin-V-7AAD-negative cells, was scored using a flow cytometer.

**Mitochondrial membrane potential assays for AML cells**. Mitochondrial membrane potentials (ΔΨm) were determined by flow cytometry using tetra-methylrhodamine ethyl ester (TMRE) as the fluorescent probe. KG1 and HL60 cells ($2.5 \times 10^5$/ml) were treated with increasing concentration of DDA for 24 h. Cells were then harvested, washed with PBS, resuspended in 1 ml of PBS, and incubated with 10 nM TMRE for 30 min. ΔΨm were analyzed for fluorescence intensity by using flow cytometry. To determine superoxide anion generation, cells were kept at 37 °C for 15 min in the presence of 10 mM HE (Molecular Probes). After incubation with DiOC6(3) or HE, the cells were immediately analyzed by flow cytometry on a BD Facscalibur flow cytometer (BD Biosciences). DiOC6(3) was excited at 488 nm and detected at 525 nm. HE was excited at 490 nm and detected at 620 nm. The median fluorescence intensity was quantified by FlowJo software.

**Histology and immunohistochemistry**. Tumors or bones were fixed in 10% neutral buffered formalin and embedded in paraffin. Paraffin sections were stained with hematoxylin and eosin or with Goldner's Trichrome as indicated for histo-morphological analyses. Immunohistochemical staining was performed on paraffin-embedded tissue sections using the following antibodies: anti-LC3 mouse (NanoTools, clone 5F10), anti-Nur77 goat (LS-Bio), anti-NOR1 rabbit (LS-Bio), and anti-P62 guinea pig (Progen Biotechnik, Heidelberg, Germany). Immunostaining was preceded by an antigen retrieval technique by heating in Tris/EDTA buffer pH 9 in a water-bath at 95 °C for 45 min. After incubation with primary antibody for 2 h at room temperature, sections were incubated with the Dako EnVision-peroxidase system (Dako S.A.S., Trappes, France) for LC3 or with a biotin-conjugated secondary Dako antibody followed by the streptavidin-biotin-peroxidase complex (Vectastain ABC kit, Vector Laboratories, CA) and then were counter-stained with hematoxylin.

**Measure of DDA efficacy in vivo**. Exponentially growing cells were harvested, washed twice in PBS then resuspended in PBS. SKMEL-28, shSKMEL-28 clones ($5 \times 10^6$ cells) and B16F10 ($3 \times 10^5$) cells were injected subcutaneously (s.c.) into the flank of nude mice. HL60, KG1 and shKG1-clones ($5 \times 10^6$) were inoculated s.c. into the flanks of NOD/SCID mice. When tumors were palpable, animals ($n = $ 10–20 mice per group, as indicated) were randomized to receive either DDA (20 mg/kg) intraperitoneally (i.p.) or 40 mg/kg by oral gavage (p.o), once a day, or vehicle solvent for the indicated time. Animals were examined daily and body weights were measured twice a week. In all experiments, tumor volume was determined by direct measurement with a caliper and was calculated using the formula (width$^2 \times$ length)/2. At the end of experiments, mice were sacrificed and tumors were excised and divided for intra-tumoral DDA quantification, immunoblotting, or immunohistochemistry analyses.

**AML patient and normal bone marrow samples**. AML patient and normal bone marrow samples were obtained from patients in the Hematology Department of Toulouse (France), after obtaining consent in accordance with the Declaration of Helsinki[67], and according to institutional guidelines. Samples were stored at the INSERM-1037 HIMIP collection (no. DC-2008-307-CPTP1 HIMIP). For some experiments, fresh leukemic blasts recovered at diagnosis were immediately treated with DDA. In other cases, frozen cells were thawed in IMDM medium with 20% FBS. Patient characteristics are shown in supplementary data 1. Normal bone marrow cells were subjected to Ficoll-Paque density gradient separation to isolate mononuclear cells, followed by CD34+ cell separation using high-gradient magnetic-activated cell sorting, according to the manufacturer's manual. The in vivo assays were performed with cryopreserved cells.

**Efficacy of DDA on human patient-derived AML**. The engraftment of human primary AML cells was performed on NSG mice. Briefly, human primary AML cells, HL60 cells or normal CD34+ cells were resuspended in PBS, and $0.8 \times 10^6$ to $2 \times 10^6$ viable trypan blue-negative cells were injected intravenously into NSG mice that had been previously irradiated for 24 h with 2.25 Gy. Two and a half weeks after the injection of human cells, mice were treated with DDA (20 mg/kg/day by IP injection) or vehicle as control for 2, 3 weeks, after which the engraftment of human AML and normal hematopoietic cells was measured in the bone marrow, spleen and brain by quantifying the percentage of hCD45+CD33+ cells by flow cytometry. The viability of AML cells was established by quantifying the percentage of hCD45+CD33+Annexin-V- cells.

**Statistical analysis**. Tumor growth curves in animals were analyzed for significance using repeated measures ANOVA. In other experiments, significant differences in the quantitative data between the control and treated groups were

analyzed using the Student's *t*-test for unpaired variables. In the figures, *, ** and *** refer to $P < 0.05$, $P < 0.01$, and $P < 0.001$, respectively, compared with controls (vehicle), unless otherwise specified. Prism software was used for all analyses.

**Data availability**. The data that support the findings of this study are available within the article and supplementary files, or available from the corresponding authors upon reasonable request.

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

## Acknowledgements

This work was supported by an internal grant from the "Institut National de la Recherche Médicale", the university of Toulouse III, the Agence Nationale pour la Recherche (ANR-11-RPIB-015-02 DAML, ANR-11-PHUC-0001, ANR-10-LABX-57); The "Ministère de la jeunesse, de l'éducation nationale et de la Recherche" (GenHomme 03L152); Onco San Tech (projet DEMODA RMN13001BBA). G.S. was supported by the Ministère Français de la Recherche and by the Association pour la Recherche sur le Cancer (DOC20110602926).

## Author contributions

Conception and design: S.S.-P., C.R. and M.P. Acquisition of data G.S., M.D., P.d.M., M.C.P., A.M., N.S., F.V., E.S., K.C., J.L., N.C., M.V., J.C., L.L., F.L., E.N., A.R., B.P., T.A.S., A.L., G.D., J.-M.L., S.B., C.D., F.d.T., C.L., H.B., F.T., M.T., D.P., M.P. and S.S.-P. Analysis and interpretation of data: G.S., M.D., P.d.M., M.R., C.R., M.P. and S.S.-P. Writing, review, and/or revision of the manuscript S.S.-P., G.S., C.R. and M.P. Administrative, technical, or material support D.P., J.-M.L., J.-E.S. and S.B. Study supervision S.S.-P., C.R. and M.P.

## Additional information

**Competing interests:** P.d.M., A.R., N.C. and E.N. are employed by and S.S.-P. and M.P. are founders of the company Affichem. The remaining authors declare no competing financial interests.

