## [Peer Review File · Nature Communications]

Reviewers' comments:

Reviewer #1 (Remarks to the Author):

In this paper the authors extend their previous work on the anti-tumor properties of the oxysterol derivative, DDA, by showing that this activity requires LXRbeta and the NR4A nuclear receptors. They also extend the anti-cancer effects to AML. Given that the authors have already established the anti-cancer effects of DDA in their previous 2013 paper (Nature Comm. DOI: 10.1038/ncomms2835), the novelty of the present work resides in the demonstration that LXRbeta is apparently the target. The data presented generally support the conclusion that LXRbeta is involved (this part of the paper is very good), but less clear are the experiments demonstrating LXRbeta is the primary therapeutic target.

Specific points:

In their 2013 paper, the authors concluded that DDA likely works by blocking LXR activation through its effects on cholesterol epoxide hydrolase, which DDA potently inhibits. In this paper they now reveal that DDA is also a high affinity LXRalpha and LXRbeta ligand, but that as such it only has partial agonist activity, and this agonist (not antagonist) activity is required for the anti-tumor effects through its ability to induce NR4A gene expression. The authors use shRNAs against the two LXRs to show that LXRbeta but not alpha is required in cell lines for these effects of DDA. However, there are several caveats to the notion that LXRbeta is the primary target. A number of important unanswered questions remain:

1. The mechanism of the specificity of DDA as a partial, gene specific, and LXR isoform specific ligand was not addressed. This is the crucial missing link in the paper and without it, the conclusions are not convincing. It is suspicious that DDA binds both LXRalpha and beta with equal affinity and equally inhibits reporter gene activity, but as an anticancer agent it only appears to work via LXRbeta and this is an exclusive effect of inducing NR4A gene expression. It is further surprising that none of the other many synthetic and natural LXR ligands are able to mimic this effect. In fact, the effect appears to be so selective that it suggests an unprecedented mechanism in nuclear receptor signaling. Does the mechanism work through promoter specific recruitment of cofactors? If so, how is this accomplished?
2. There are no data showing LXRbeta (and not alpha) is selectively recruited to Nur77, Nor1, and the L3C gene promoters. Nor is there any evidence to show there are functional response elements in these promoters. The ChIP-seq data are only suggestive.
3. One prediction of the model is that other synthetic LXR agonists should potently inhibit the in vitro and in vivo effects of DDA via competitive binding to LXRbeta. Is this the case?
4. It is not clear what the role of the ChEH enzyme is in the mechanism, particularly when one takes into account the authors' previous paper. The authors did one set of cursory experiments to suggest that the enzyme activity was not required in the cell lines (Fig 1C), but this was not convincing in completely ruling out that it has a role on the in vivo anti-tumor activity of DDA.
5. The requirement for NR4A receptors was not entirely convincing. The KD of these was more than 90% but the effects on DDA inhibition were roughly only 50%.

Reviewer #2 (Remarks to the Author):

This is a very interesting study in which the authors demonstrate that dendrogenin A (DDA), a new cholesterol metabolite with tumor suppressor properties, promotes lethal autophagy through its role as a partial agonist of the nuclear receptor LXRbeta. They show that DDA, through LXRbeta, increased the levels of Nur77, Nor1 and LC3, as well as that it triggered the formation of autolysosomes. The authors also showed that DDA inhibited the cholesterol biosynthetic enzyme D8D71, which results in sterol accumulation that might cooperate in the promotion of autophagy. The experiments reported in this manuscript seem to have been performed competently and with statistical rigor. The results are novel and of potential significance for our understanding of the role

of autophagy not only in cell survival but also in cell death. This latter is an area poorly explored in the autophagy field and with potential therapeutic value in cancer. The only weakness of the study is the lack of any results and/or discussion on the molecular mechanisms that establish the decision point between survival autophagy vs. lethal autophagy. In other words, what makes DDA-LXRbeta to promote a form of autophagy that results in cell death instead of survival. This important question should have been, at a minimum, mentioned in great detail in the Discussion, and put in context with the vast amount of data available supporting autophagy as a cell survival and tumor promoting activity.

Point by point rebuttal:

Reviewers' comments:

Reviewer #1 (Remarks to the Author):

In this paper the authors extend their previous work on the anti-tumor properties of the oxysterol derivative, DDA, by showing that this activity requires LXRbeta and the NR4A nuclear receptors. They also extend the anti-cancer effects to AML. Given that the authors have already established the anti-cancer effects of DDA in their previous 2013 paper (Nature Comm. DOI: 10.1038/ncomms2835), the novelty of the present work resides in the demonstration that LXRbeta is apparently the target. The data presented generally support the conclusion that LXRbeta is involved (this part of the paper is very good), but less clear are the experiments demonstrating LXRbeta is the primary therapeutic target.

Specific points:

In their 2013 paper, the authors concluded that DDA likely works by blocking LXR activation through its effects on cholesterol epoxide hydrolase, which DDA potently inhibits.

Answer :

We are sorry, but in our Nature Commun 2013 paper, we did not work on the LXRs, no experiments have been done on the LXRs. We have only evaluated the impact of DDA on the ChEH. We did not “conclude” that DDA works by blocking the LXRs through the ChEH. In fact, we gave only hypotheses in the discussion of how DDA at low doses may induce the infiltration of the immune system on syngeneic mouse tumors implanted into immunocompetent mice, and the hypothesis was that DDA could interact indirectly through the LXR by producing sulfated 5,6EC which are LXR ligands, since LXRs were described to modulate the immune system. Moreover, at that time, we had no evidence that DDA could control the growth of human tumors implanted into mice and its mechanism of cytotoxicity. The objective of the present paper was to evaluate DDA against human tumors implanted into immunodeficient mice, melanoma and leukemia, including patient samples and to determine whether DDA could have a direct anti-tumor effect on human tumors with a deficient immune system. Thus the results, we have obtained on human tumors are original and were not predictable from our previous published works and shed light on how this molecule works and gives new important data on the field. Specifically it describes for the first time that LXR can control autophagic programs at the transcriptional level.

This is the text in the conclusion the reviewer referred to:

“Oxysterol metabolism has been linked to innate and adaptive immune responses through LXR signalling^{40,41}, and 5,6 α -EC was reported to be a direct modulator of LXR¹⁷. In addition, 5,6 α -EC is the substrate of cholesterol sulphotransferase SULT2B1b^{42,43}, and the resulting 5,6 α -EC-3-sulphate is a modulator of LXR^{9,11,18}. A recent study showed that human and mouse tumours produce LXR agonists that inhibit CC chemokine receptor-7 expression on maturing DCs and the ability of DCs to initiate an immune T-cell response against tumours. Engineering the

expression of SULT2B1b in these tumours reversed this process through inactivation of LXR agonists and resulted in tumour growth control, increased animal survival and inhibition of tumour immunoescape⁴⁰. This work shed light on the importance of regulating LXR to mediate an anti-tumour response, immunity and animal survival. This could be achieved with DDA through the direct inhibition of ChEH, which generates LXR modulators^{9,11,18}. “

Reviewer :

In this paper they now reveal that DDA is also a high affinity LXRalpha and LXRbeta ligand, but that as such it only has partial agonist activity, and this agonist (not antagonist) activity is required for the anti-tumor effects through its ability to induce NR4A gene expression. The authors use shRNAs against the two LXRs to show that LXRbeta but not alpha is required in cell lines for these effects of DDA. However, there are several caveats to the notion that LXRbeta is the primary target. A number of important unanswered questions remain:

1. The mechanism of the specificity of DDA as a partial, gene specific, and LXR isoform specific ligand was not addressed. This is the crucial missing link in the paper and without it, the conclusions are not convincing. It is suspicious that DDA binds both LXRalpha and beta with equal affinity and equally inhibits reporter gene activity, but as an anticancer agent it only appears to work via LXRbeta and this is an exclusive effect of inducing NR4A gene expression.

It is further surprising that none of the other many synthetic and natural LXR ligands are able to mimic this effect. In fact, the effect appears to be so selective that it suggests an unprecedented mechanism in nuclear receptor signaling. Does the mechanism work through promoter specific recruitment of cofactors? If so, how is this accomplished?

We apologize for the confusion but we haven't focused on the LXR α because the LXR α was not expressed in melanoma cells or its expression was 10 time lower compared to the LXR β in AML cells (see Figure S5M), and thus the main isoform expressed in the models studied was the LXR β . This differential expression in favor of LXR β was already reported in the literature in the NCI60 cancer cell panel (Holbeck S et al, Mol Endo, 2010) including melanoma and AML tumor cells as well as in patient tumors (Pencheva N et al, cell 2014; Huffman KE et al, Front endocrinol, 2015). This is also observed in other tumors such as glioblastoma (Villa, GR. et al, Cancer cell, 30, 683–693).

Similarly, we found that the LXR β (NR1H2) is the predominant LXR isotype over LXR α (NR1H3) in metastatic melanoma (SKCM, 473 patients) and in AML (LAML, 173 patients), two cohort of patients analyzed by mRNA sequencing at The Cancer Genome ATLAS (TCGA), now included in Figure 4 J and in Figure 6L

When we did the invalidation of the LXR α gene expression in AML cells, we observed no measurable impact on the cytotoxicity induced by DDA. The 10-fold lower expression of LXR α

versus that of LXR β (see Figure S6M) may explain this result by making that the loss of LXR α may be compensated by the LXR β that has greater expression and inversely making that the loss of the LXR β cannot be compensated by the LXR α . We have now discussed these considerations in the manuscript p10: “LXR α and LXR β are expressed in the two tested AML cell lines, but LXR α was weakly expressed compared to LXR β (Fig.S6M). Their KD (Fig.S6N and Fig.5I) showed that LXR β mediated DDA-induced cell death (Fig.5J and S6O) and autophagy (Fig.5K and S6P-Q) in these cells, which does not rule out a possible contribution of LXR α .” What we claimed here is that LXR β , the main isoform expressed in melanoma and AML cells, induces the anti-tumor activity of DDA and is involved in the control of DDA-induced cytotoxicity in these cells. To avoid confusion, we propose to change the title by replacing LXR β by LXR.

Moreover, our results clearly indicate that DDA binds to the two isoforms (FIG 3B and 3C) and inhibits the transcriptional activity of 22(R)HC (Fig 3A) on the two isoforms and the structure/transcriptional activity studies performed on DDA on the two isotypes also confirmed that DDA is not a selective ligand of LXR α or LXR β (Fig 3F). Thus, the effect of DDA on tumor cell we studied reflect mostly its effect through the LXR β because it is the main or the only isoform expressed in these models. As we said above, the LXR β is the predominant isoform in melanoma and AML cells.

We have added a control Surface Plasmon Resonance experiment with GW3965 on Fig.S3H.

The text was modified as followed p 6:

“Binding competition assay (Fig.3B) and surface plasmon resonance assays (Fig.3C) on recombinant LXR α and LXR β ligand binding domains (LBD) indicated that DDA is a ligand of both isoforms with a 4-fold preference for LXR β -LBD, while GW3965 displayed a similar affinity for both LXR-LBD (Fig.S3H).”

2. There are no data showing LXR β (and not alpha) is selectively recruited to Nurr77, Nor1, and the L3C gene promoters. Nor is there any evidence to show there are functional response elements in these promoters. The ChIP-seq data are only suggestive.

Answer :

Since the LXR β is the predominant form expressed in melanoma and AML cells, we assessed whether DDA affects the binding of LXR β on the potent enhancers identified in Fig.S4G. DDA increased the binding of LXR β on enhancers that are close to well known target genes of LXR β (SCD1, SREBF1, Fig.A) while it repressed LXR β binding on other enhancers (ABCA1, LDLR, Fig.A), showing that DDA directly regulates LXR β target genes. LXR β binding on enhancers close to autophagic genes (LC3s and NR4As) is globally increased by DDA (Fig.3I) suggesting a direct control of their expression by DDA through LXR β . 22(R)HC also regulate the binding of LXR β on these enhancers reinforcing the possibility that LC3s and NR4As are LXR β target

genes. Importantly, differences appeared between DDA and 22(R)HC regarding the binding of LXR β , in a gene-specific manner (Fig.3I). This suggests that a set of LXR β target genes is differently controlled by DDA compared to other LXR ligands and may explain why the anticancer mechanisms induced by DDA were never reported until now for LXRs.

To gain insights into target genes of LXR β that might be involved in both processes of autophagy and lysosome biogenesis, we computationally predicted potent target genes of LXR β based on the proximity between their Transcription Start Site (TSS) and LXR β binding sites (LXRBS) (Wang S, Nature Protocols 2013). Then we performed a gene ontology (GO) study on these putative LXR β target genes and we only found TFEB in both « Autophagy » and « Lysosome organization » GO terms (Fig.S4H).

We confirmed by ChIP that TFEB is a direct target gene of LXR β (Fig.3J) because both 22(R)HC and DDA controlled the binding of LXR β on an enhancer located 5.7 kilobases from the TSS of TFEB. While DDA decrease the binding of LXR β on TFEB enhancer and stimulates its expression in a dose-dependent manner, 22(R)HC did exactly the opposite (Fig.3J), suggesting that LXR β and LXR β agonists act as repressors of TFEB and that DDA derepressed it. This may explain why some effects of DDA on lysosome biogenesis cannot be affected by knocking down LXR β . Finally, the activation of TFEB expression by DDA increased the activity of a TFEB-dependent gene reporter assay (Fig.3K) in a dose-dependent manner showing that DDA stimulated the transcriptional activity of TFEB. As TFEB is a master transcription factor controlling genes involved in autophagy and lysosome biogenesis, our results suggest that part of autophagy induction by DDA is mediated through the derepression of TFEB by the direct inhibition of LXR β binding, in addition to the direct stimulation of the expression of NR4As genes by LXR β .

The text page 8 was modified as followed:

“We next assessed whether DDA affects the binding of LXR β on the potent enhancers identified in Fig.S4G. DDA increased the binding of LXR β on enhancers that are close to well-known target genes of LXR β (SCD1, SREBF1), while it repressed LXR β binding on other enhancers (ABCA1, LDLR), showing that DDA directly regulates LXR β target genes (Fig 4I). LXR β

binding on enhancers close to autophagic genes (LC3s and NR4As) is globally increased by DDA (Fig 4I) strongly suggesting a direct control of their expression by DDA through LXR β . 22(R)HC also regulates the binding of LXR β on these enhancers reinforcing the possibility that LC3s and NR4As were LXR β target genes. Importantly, differences appeared between DDA and 22(R)HC regarding the binding of LXR β , in a gene-specific manner (Fig.3I). This suggests that a set of LXR β target genes is differently controlled by DDA compared to other LXR ligands and may explain why the anticancer mechanisms induced by DDA were never observed until now with LXR ligands. To gain insights into LXR β target genes that might be involved in autophagy and lysosome biogenesis, we computationally predicted putative LXR β -target genes based on the proximity between their Transcription Start Site (TSS) and LXR β binding sites (LXRBS) according Wang et al methodology²⁷. We performed a gene ontology (GO) study using « Autophagy » and « Lysosome organization » terms on these putative LXR β target genes and got 26 hits including MAP1LC3B that could be regulated by LXR. Among them, TFEB was common to both series (Fig.S4G). CHIP analyses of 22(R)HC and DDA treated SKMEL-28 cells showed that they increased or inhibit respectively LXR β binding to an enhancer located 5.7 kilobases from the TSS of TFEB (Fig.3J) confirming that TFEB was a direct LXR β -target gene. DDA decreased LXR β -binding to a TFEB enhancer and stimulates TFEB expression dose-dependently (Fig.3K), 22(R)HC did the opposite, suggesting that LXR β and LXR β agonists act as repressors of TFEB and that DDA derepressed this gene. Finally, the activation of TFEB expression by DDA increased the activity of a TFEB-dependent gene reporter assay (Fig. 3L) dose-dependently showing that DDA stimulated the transcriptional activity of TFEB. TFEB is a master transcription factor controlling genes involved in autophagy and lysosome organization and biogenesis^{28, 29, 30, 31}. These data revealed that LXR through DDA binding appears as a new player in autophagy and lysosomes biogenesis. Our results suggest that part of autophagy induction by DDA could be mediated through the derepression of TFEB by the direct inhibition of LXR β binding, in addition to the direct stimulation of the expression of NR4As genes by LXR β .

3. One prediction of the model is that other synthetic LXR agonists should potently inhibit the in vitro and in vivo effects of DDA via competitive binding to LXRbeta. Is this the case?

Answer :

To answer the this point we have performed experiments in vitro and in vivo:

In vitro we showed that treatment of cells with 5 μ M 22RHC, 0.5 μ M TO or 1 μ M GW protected cells from DDA induced cytotoxicity at 2 μ M. We next looked at the impact of treatment of cells with TO on DDA induced autophagosome and autolysosome formation and found that TO protected against the stimulation induced by DDA.

The text was modified as followed:

“In contrast, 22(R)HC, TO and GW had no impact on cell death (Fig.3O), consistent with work from Pencheva *et al.*³² and partially reversed DDA cytotoxicity (Fig.3O). As expected, we found that TO decreased the stimulation of autophagic vesicles formation by DDA (Fig.3P).”

In vivo experiments were conducted on B16F10 and SKMEL-28 cells implanted on mice. 20mg/kg TO + 20mg/kg DDA was administered (figure S4I)

We found that TO was able to reverse the growth control induced by DDA. This further support the importance of LXR in the induction of cell cytotoxicity and the control of the growth of tumors in vivo.

The text was modified as followed p10:

“We found that TO partially reversed the inhibition of tumor growth induced by DDA on melanoma cells (Fig S4I). Analyses of the sterol profile in tumors confirmed the accumulation of $\Delta 8$ -sterols in tumors, while no increase in 5,6-EC was measured (Fig S4I) as observed on *in vitro* tests (Fig.1D and Fig.S2A). LXR β is the predominant isotype expressed in melanoma cells^{32, 33, 34, 35} and analyses of the TCGA dataset expression (Fig.4J) corroborate these observations and supporting the importance of LXR β targeting in melanoma cells.”

4. It is not clear what the role of the ChEH enzyme is in the mechanism, particularly when one takes into account the authors’ previous paper. The authors did one set of cursory experiments to suggest that the enzyme activity was not required in the cell lines (Fig 1C), but this was not convincing in completely ruling out that it has a role on the in vivo anti-tumor activity of DDA.

We have added two sets of experiments to address this question. If ChEH is involved on DDA activity, than we might see an increase on cholesterol epoxides under DDA treatment either on cell culture or on tumors. We found no increase on 5,6-EC (Fig S2A) which rules out 5,6-EC metabolism on DDA activity. Comparison with other ChEH inhibitors Tamoxifen and PBPE showed that they increase 5,6-EC production and accumulation in B16F10 and SKMEL-28 cells (Fig S2A). The explanation is that DDA, but not PBPE or Tam, stimulates the expression of catalase, which destroys H₂O₂ and thus does not trigger cholesterol epoxidation (Fig S2B).

We next performed dosage of $\Delta 8$ -sterols and 5,6EC on tumors implanted on mice treated or not with DDA, GW, DDA + GW. We measured an increase in $\Delta 8$ -sterols in tumors when

mice were treated with DDA and DDA + GW but not with GW alone (Fig S4I). We found no increase in 5,6-EC accumulation in tumors under DDA treatment but rather a significant decrease and TO blocked this decrease.

These data ruled out the implication of ChEH activity in DDA action in vivo and highlight the implication of its D8D7I subunit on DDA action in vivo.

We added a series of schemes (Fig S2C-G) to explain the differences between DDA and ChEH inhibitors we noticed:

The following text was added page 4 and 9:

“Analyses of the oxysterol profile of cells treated with DDA showed no accumulation in 5,6-EC as opposed to what was found with other ChEH inhibitors Tam and PBPE (Fig S2A). We observed that DDA stimulated catalase activity (Fig S2B), which induced H₂O₂ destruction and blocked 5,6-EC production. This established a significant difference between DDA and ChEH inhibitors like Tam or PBPE (Fig S2C-D) because we showed that Tam and PBPE mediated part of their cytotoxicity through the accumulation of 5,6 α -EC, which acted as a second messenger¹⁹.”

“Analyses of the sterol profile in tumors confirmed the accumulation of Δ 8-sterols in tumors, while no increase in 5,6-EC was measured (Fig S4I) as observed on *in vitro* tests (Fig.1D and Fig.S2A).”

5. The requirement for NR4A receptors was not entirely convincing. The KD of these was more than 90% but the effects on DDA inhibition were roughly only 50%.

Answer :

The 10% left for each NR4A receptor seems to be sufficient to mediate 50 % cytotoxicity. This is observed for some receptors for which some pool of receptors are in reserve. Moreover, other actors of autophagy also contribute to DDA-induced cell death such as LC3, other ATG proteins, TFEB and Beclin 1

We agree with the reviewer and we think that the supplementary data we provided on the revised version of our manuscript in particular with ChIP analyses and computational analyses support the implication of LXR in the activation of LXR-dependent autophagic programs by DDA.

Reviewer #2 (Remarks to the Author):

This is a very interesting study in which the authors demonstrate that dendrogenin A (DDA), a new cholesterol metabolite with tumor suppressor properties, promotes lethal autophagy through its role as a partial agonist of the nuclear receptor LXRbeta. They show that DDA, through LXRbeta, increased the levels of Nur77, Nor1 and LC3, as well as that it triggered the formation of autolysosomes. The authors also showed that DDA inhibited the cholesterol biosynthetic enzyme D8D7I, which results in sterol accumulation that might cooperate in the promotion of autophagy. The experiments reported in this manuscript seem to have been performed competently and with statistical rigor. The results are novel and of potential significance for our understanding of the role of autophagy not only in cell survival but also in cell death. This latter is an area poorly explored in the autophagy field and with potential therapeutic value in cancer. The only weakness of the study is the lack of any results and/or discussion on the molecular mechanisms that establish the decision point between survival autophagy vs. lethal autophagy. In other words, what makes DDA-LXRbeta to promote a form of autophagy that results in cell death instead of survival. This important question should have been, at a minimum, mentioned in great detail in the Discussion, and put in context with the vast amount of data available supporting autophagy as a cell survival and tumor promoting activity.

Answer :

We have now discussed this important point as followed p14

“In addition, our study revealed a yet unprecedentedly described relationship between LXR and the control of the expression of genes involved in autophagy and lysosomes biogenesis.

Autophagy is a well established survival mechanism that helps cells to escape to death and is also a mechanism of resistance to chemotherapy and endocrine therapies^{18, 19, 44, 45, 46}. ChEH inhibitors Tam and PBPE stimulate a protective autophagy in melanoma cells and other cancer cells through D8D7I inhibition and $\Delta 8$ -sterols accumulation^{10, 11, 13, 19}. We report here that DDA mechanisms are different from these compounds. DDA inhibits D8D7I leading to $\Delta 8$ -sterols accumulation and additionally binds to LXR and stimulate the expression of pro-autophagic genes. Thus the activation of a multiple autophagic regulatory circuitry through new mechanisms with a single molecule may account for lethal autophagy as observed with some combination therapy approaches^{18, 47}.”

REVIEWERS' COMMENTS:

Reviewer #1 (Remarks to the Author):

The authors have addressed my concerns.

Reviewer #2 (Remarks to the Author):

The authors have satisfactorily addressed my previous concern.

Answer to reviewers:

REVIEWERS' COMMENTS:

Reviewer #1 (Remarks to the Author):

The authors have addressed my concerns.

Reviewer #2 (Remarks to the Author):

The authors have satisfactorily addressed my previous concern.

The authors thank the reviewers for their positive response